# Demultiplexing and barcode-specific adaptive sampling for nanopore direct RNA sequencing

Wiep van der Toorn [1,2,7], Patrick Bohn [3,7], Wang Liu-Wei [1,2,4], Marco Olguin-Nava[3], Anne-Sophie Gribling-Burrer[3,5], Redmond P. Smyth [3,5,6] ✉ & Max von Kleist [1,2] ✉

Nanopore direct RNA sequencing (dRNA-seq) enables unique insights into RNA biology. However, applications are currently limited by the lack of accurate and cost-effective sample multiplexing. Here we introduce WarpDemuX, an ultra-fast and highly accurate adapter-barcoding and demultiplexing approach for dRNA-seq with SQK-RNA002 and SQK-RNA004 chemistries. WarpDemuX enhances speed and accuracy by fast processing of the raw nanopore signal, use of a light-weight machine-learning algorithm and design of optimized barcode sets. We demonstrate its utility by performing rapid phenotypic profiling of different SARS-CoV-2 viruses through multiplexed sequencing of longitudinal samples on a single flowcell, identifying systematic differences in transcript abundance and poly(A) tail lengths during infection. Additionally, integrating WarpDemuX into sequencing control software enables real-time enrichment of target molecules through barcode-specific adaptive sampling, which we demonstrate by enriching low abundance viral RNA. In summary, WarpDemuX represents a broadly applicable, high-performance, economical multiplexing solution for dRNA-seq, facilitating advanced (epi-) transcriptomic research.

Nanopore direct RNA sequencing (dRNA-seq) is a powerful tool for transcriptomics[1]. Unlike other RNA sequencing methods, dRNA-seq does not require the conversion of RNA to cDNA. It therefore allows for the direct identification of RNA isoforms[2] and associated transcript features such as polyA tail length[3]. This not only provides a more comprehensive view of the transcriptome, it also preserves native RNA base modifications, offering a unique opportunity to study the dynamic epitranscriptome[4].

A common sequencing practice known as multiplexing allows different samples to be pooled and sequenced simultaneously. Multiplexing reduces costs while minimizing experimental variability and is essential for modern sequencing-based assays, such as single cell sequencing[5]. Multiplexing involves the attachment of sample specific barcodes to each sequencing library before pooling. This allows the re-identification of sample-specific sequencing data for subsequent analysis. While multiplexing protocols for nanopore DNA-sequencing (DNA-seq) are available, no commercial multiplexing protocols have been released for dRNA-seq.

Multiplexing in nanopore sequencing is challenging due to its unique methodology, where individual molecules are threaded through a protein (nano-) pore that is embedded in an electrically resistant polymer membrane. During pore translocation, features of

[1]Systems Medicine of Infectious Disease (P5), Robert Koch Institute, Berlin, Germany. [2]Department of Mathematics and Computer Science, Freie Universität Berlin, Berlin, Germany. [3]Helmholtz Institute for RNA-based Infection Research, Helmholtz Centre for Infection Research, Würzburg, Germany. [4]International Max-Planck Research School for Biology and Computing (IMPRS-BAC), Berlin, Germany. [5]Architecture et Réactivité de l'ARN, Institut de biologie moléculaire et cellulaire du CNRS, Université de Strasbourg, UPR 9002, Strasbourg, France. [6]Department of Medicine, University of Würzburg, Würzburg, Germany. [7]These authors contributed equally: Wiep van der Toorn, Patrick Bohn. ✉e-mail: redmond.smyth@helmholtz-hiri.de; max.kleist@fu-berlin.de

the molecule cause systematic disruptions to an ionic current flow. Controlled movement is achieved using a motor protein associated with a DNA sequencing adapter that is ligated to the end of the molecule during library preparation. This leads to step-wise changes in electrical signals that are characteristic of a group of nucleotides (*k-mer*) passing through the pore at a given time[6]. Repeated measurements are made at each translocation event, and the number of repeated measurements per event is commonly termed the *dwell time*. Intrinsic fluctuation of dwell time is a characteristic of nanopore sequencing, and the resulting temporal variation can pose challenges in accurately converting the signal into a nucleotide sequence (*basecalling*).

For DNA-seq, commercial protocols incorporate the specific nucleotide barcode within the DNA sequencing adapter, which is later read out natively by the DNA basecaller (basecalling-based demultiplexing). dRNA-seq uses similar DNA sequencing adapters to attach the motor protein during library preparation, resulting in a hybrid (DNA-RNA) molecule. Because the DNA portion of the molecule cannot be natively basecalled by the RNA-specific basecaller, it is not possible to read out the specific nucleotide sequences of barcodes. Conceptually, this problem can be overcome by ligating RNA barcodes to the 3' end of the transcript. However, RNA barcodes require additional ligation and purification steps, which increases the complexity and duration of the library preparation process, and potentially introduces ligation biases. Moreover, the presence of the DNA sequencing adapter adjacent to the RNA barcode negatively affects the barcode basecall quality, complicating demultiplexing. Addressing this with linkers is undesirable due to high costs: RNA oligos are approximately 10-fold more expensive than DNA oligos (at 6€ per nucleotide, ordering 12 16-mer barcodes amounts to roughly 1200€)

Alternatively, barcode sequences can be incorporated into the DNA adapter and demultiplexed from their raw electrical signals (signal-based demultiplexing). The DNA adapter (RTA) contains an 18-nucleotide stretch that can be customized to include a barcode, and its ligation to the 3' end of the RNA is integral to dRNA-seq library preparation, thus barcoding would not complicate the experimental protocol. Signal-based demultiplexing was first realized in a method named DeePlexiCon (DPC)[7]. Its main algorithm transforms the raw adapter signal to images, which are subsequently classified using a deep convolutional neural network. Unfortunately, this approach requires extensive computational resources, such as GPUs, to achieve reasonable run times. Despite only using four barcodes, intrinsic variation in the dwell time produces inconsistent signal-transformed images leading to substantial misclassification.

Here, we developed WarpDemuX, an efficient and scalable method to design and detect RTA barcodes for nanopore dRNA-seq. WarpDemuX operates directly on the raw signal and does not require basecalling. It uses effective signal preprocessing and a fast machine-learning algorithm for barcode classification. We obtain an optimized set of 12 RTA barcodes with maximized inter-barcode distances in the signal space for high-accuracy demultiplexing. We also establish real-time demultiplexing and barcode-specific adaptive sampling[8] for dRNA-seq.

## Results

### High demultiplexing accuracy, yield and speed using WarpDemuX

We aimed to establish a robust, low-cost, and accurate barcoding methodology to enable dRNA-seq multiplexing and adaptive sampling experiments. We chose to place the barcode sequences within the customizable RTA adapter due to its compatibility with the dRNA-seq library preparation workflow. Because the DNA RTA adapter cannot be read by the RNA-specific basecaller, we implemented signal-based demultiplexing.

Dynamic Time Warping (DTW) is an algorithm used for comparing time-series with temporal distortions. This makes it ideally suited for analyzing raw nanopore signals with their intrinsic variations in dwell-time[9–12]. To apply this algorithm to dRNA-seq multiplexing, we integrated a Dynamic Time Warping Distance (DTWD) kernel into a Support Vector Machine (SVM) classifier. This DTWD-based kernel function captures the essential spatial and temporal signal information by quantifying how similar an unknown barcode is to known barcode patterns, termed *fingerprints*.

For training the SVM-DTWD model, fingerprints were obtained by sequencing dual-barcoded molecules. These were generated by ligation of known RTA barcodes to distinct in vitro transcribed RNA molecules with an integrated in-line RNA barcode (see Methods for more details) (Fig. 1a). Unique combinations of RNA transcripts, in-line RNA barcodes and RTA barcodes across replicates were used to establish a ground truth label for the RTA barcode based on the basecalled sequence of the RNA transcript and RNA barcode (Supplementary Table 1–3) to provide a more reliable approach than species-based genome mapping. Shuffling the assignment of in-line RNA barcodes to RTA barcodes across replicates, in addition, allowed us to evaluate performance across different RNA species and sequence contexts under real experimental conditions, testing both the technical aspects of barcode detection and the biological variability inherent in RNA sequencing. To extract the fingerprints, we first identified the adapter signal by locating the DNA/RNA boundary in the raw signal (Fig. 1b). We then segmented the adapter signal into discrete sequencing events (Fig. 1c). Finally, we extracted the last 25 segments, expected to contain the barcode signal (Fig. 1a–c).

In our barcode classification approach (Fig. 1d), a confidence score is computed to assess the reliability of each classification result (see Methods). By setting a confidence threshold and discarding predictions below it as unclassified, users can balance accuracy and yield according to their needs. It is important to note that the fraction of unclassified reads is influenced by the dataset's characteristics, not just the model's performance. Specifically, datasets with more noisy reads tend to have a reduced yield because these reads often fall below the confidence threshold, which is the desired behavior to ensure accuracy. To allow users to balance accuracy and yield independently of noise detection, we introduced a noise class into the model. This integration enhances WarpDemuX's robustness and allows for precise tuning in diverse dataset conditions. The training data for this class includes fingerprints from outlier reads affected by signal irregularities, such as stalled adapters, blocked pores, or inaccurate DNA/RNA boundaries (see Supplementary Note 1 and Supplementary Figs. 1, 2). Reads classified as noise are subsequently omitted from performance assessment. For benchmarking, we initially obtained fingerprints for four RTA barcode sequences used in DPC (Supplementary Table 4)[7]. At a confidence threshold of 0.5, WarpDemuX outperformed DPC in both demultiplexing accuracy (97% vs. 92%) and yield (unclassified 7.7% vs. 11.4%) (Table 1, Fig. 1e, f and Supplementary Figs. 3–7). This superior performance was achieved using just 1% of the training data per barcode (400 instances vs. 40,000), and WarpDemuX's enhanced classification performance over DPC held true for all confidence thresholds tested (Supplementary Table 5). Furthermore, when aiming for maximum yield by using a confidence threshold of 0, WarpDemuX still outperformed DPC (94% against 87%). Notably, in this setting, all data was assigned to barcodes rather than to the noise class. Altogether, these data argue that SVM-DTWD classification effectively captures the nuances of the barcode fingerprints and is robust to noise emanating from dwell-time variation. Importantly, this increase in accuracy was achieved with 10 times less compute time compared to DPC (14.4 ms vs. 163 ms per read on a standard laptop) (Table 1 and Supplementary Tables 6, 7). In the described set-up, WarpDemuX-DPC classifies 4000 reads in less than 1 minute, with a peak memory usage (maximum resident set size) of less than 600 MB.

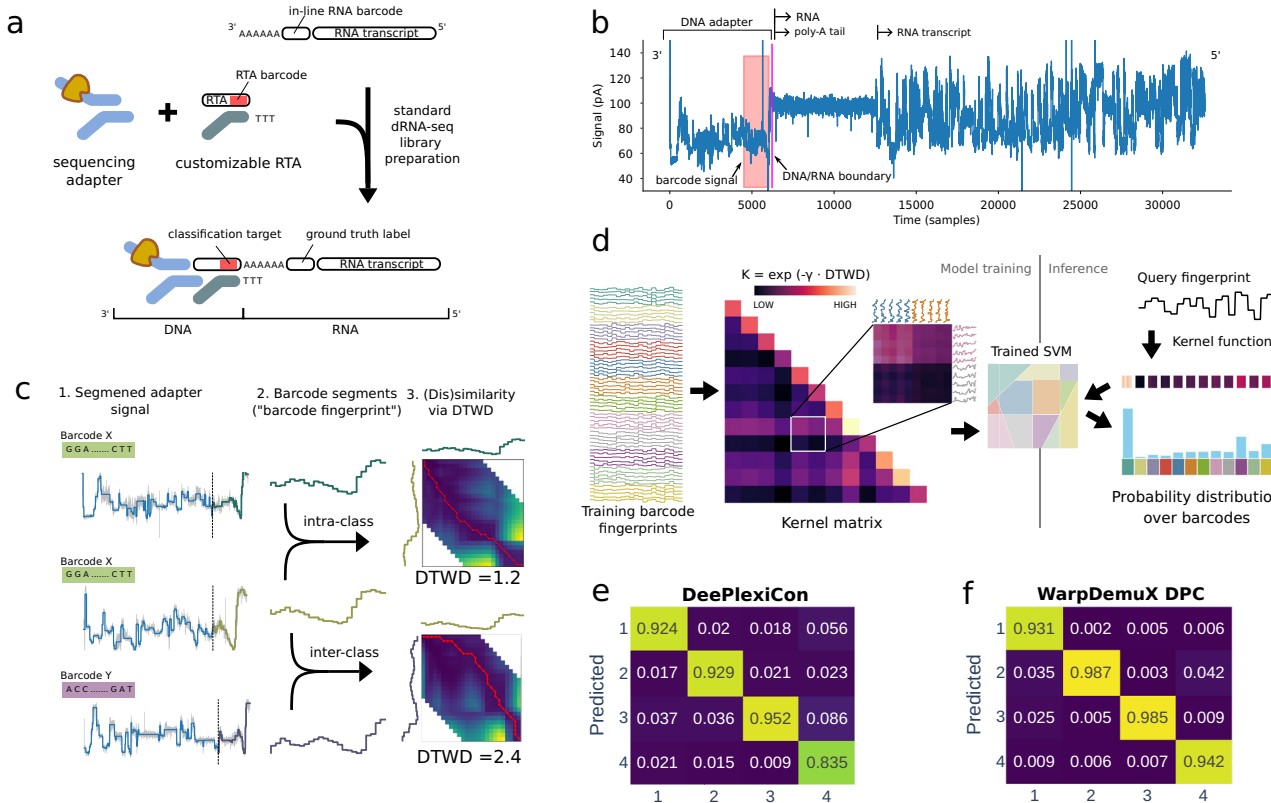

**Fig. 1 | WarpDemuX model. a** Generation of training data using dual-barcoded RNA molecules containing an in-line RNA barcode positioned at the 3′ end of the transcript, serving as a ground truth label, and an RTA barcode, introduced during standard dRNA-seq library preparation, which served as the target for classification. **b** The direct RNA sequencing raw signal denotes a time series of electric current measurement. The signal (blue) consists of three main components: the DNA adapter signal, the RNA poly(A) tail, and the RNA body. The last part of the adapter signal contains the RTA barcode to be classified (red). **c** The adapter signal is segmented into events, with each segment represented by its average signal. The final 25 segments of the adapter are designated as the RTA barcode-specific fingerprint. Dynamic Time Warping Distance (DTWD) is used to quantify the similarity between these fingerprints, with higher similarity (lower DTWD) observed within the same barcode class compared to different barcode classes. **d** Model training and classification of barcode fingerprints into barcode classes using the DTWD kernel function. Training of the Support Vector Machine (SVM) classifier with the DTWD kernel function on barcode fingerprints from sequencing reads captures the nuances of the barcode signals and promotes robust, noise-resilient classification. **e**, **f** Performance of models. Values reported reflect the recall (confusion matrix, normalized over the actual class; true label), on both test sets combined, evaluated at a confidence cutoff of 0.5. (**e**) DeePlexiCon (DPC) CNN model. **f** WarpDemuX SVM-DTWD. Source data are provided as a Source Data file.

## Optimized barcodes improve WarpDemuX performance

To expand beyond the pool of 4 barcodes, we next implemented a computational framework to obtain optimized sets of RTA barcodes with distinct signal profiles. This de novo design utilizes DTWD as a metric to maximize inter-barcode differences in signal space. In brief, we constructed a set of wave-like target signal patterns (Fig. 2a) that exhibited high inter-target DTWD. We then used the available DNA k-mer models (github.com/nanoporetech/kmer_models) to search for barcode sequences whose putative signals would closely match these target patterns (see Supplementary Method 1). From this reduced number of candidate barcodes, we selected the set of $k$ (here, $k = 12$) candidate barcodes that maximized all pairwise DTWDs (Supplementary Table 8). Notably, this strategy avoided an exhaustive search across all possible sets of barcodes and thereby significantly reduced the computational costs of barcode set selection (Fig. 2b) (see Methods for technical details). As before, we generated ground-truth training data for the optimized WDX-RTA barcodes using distinct dual-barcoded RNA transcripts (Supplementary Table 1).

We next assessed the performance of this design strategy by comparing non-optimized RTA barcodes (WDX-DPC) to four of the 12 optimized barcodes (WDX4) (Supplementary Note 2 and Supplementary Table 9). The use of optimized barcodes improved both accuracy (99% WDX4 vs. 97% WDX-DPC) and yield (96% WDX4 vs. 92.3%

WarpDemuX-DPC) while run times remained unaltered (Table 1 and Fig. 2c). This confirmed that maximizing DTWD improves classification performance. When the model was extended to 12 distinct barcode classes (WDX12), accuracy only slightly decreased despite the complexity of distinguishing a larger number of barcode classes (99% WDX4 vs 98% WDX12; weighted average) (Fig. 2d). A decrease in prediction confidence resulted in a slightly higher rate of unclassified barcodes (4.0% to 8.8% at a confidence threshold of 0.5) (Table 1). Remarkably, the WDX12 model consistently outperformed DPC (4 barcodes) across all confidence thresholds in accuracy, yield and run time (Fig. 2e, Table 1, Supplementary Tables 5–7 and Supplementary Tables 10–12), whilst expanding the number of barcode classes three-fold (also see Supplementary Note 3). Importantly, the performance of WarpDemuX only decreased slowly with increasing numbers of barcodes (Fig. 2f). With an optimized set of barcodes, we predict that 24 barcodes can be implemented at an accuracy of 96% with 14% of reads unclassified (confidence cutoff = 0.5) (Fig. 2f).

## Rapid phenotyping of SARS-CoV-2 viruses on a single flow cell

Rapid phenotypic profiling of viruses has critical importance in pandemic prevention and response, for which nanopore sequencing offers unique benefits[13]. Multiplexing samples on a single flow cell increases the speed and efficiency of profiling, while also reducing experimental

**Table 1 | Performance comparison between DeePlexiCon (DPC) and WarpDemuX (WDX) across datasets and barcoding strategies**

| Model | Dataset | Reads per barcode (avg) | Reads classified as noise (%) | Confidence cutoff | Unclassified reads (%) | Accuracy | Precision | Recall | Run time (ms per read) |
|---|---|---|---|---|---|---|---|---|---|
| DeePlexiCon | 1 | 488 | 0.0 | 0.5 | 11.0 | 0.925 | 0.919 | 0.907 | 163.0 |
| | 2 | 2114 | 0.0 | 0.5 | 15.9 | 0.906 | 0.859 | 0.918 | |
| WarpDemuX-DPC | 1 | 488 | 0.0 | 0.5 | 9.5 | 0.965 | 0.957 | 0.967 | 14.4 |
| | 2 | 2114 | 0.0 | 0.5 | 7.3 | 0.969 | 0.964 | 0.958 | |
| WDX4 | 3–5 (test set) | 600 | 1.8 | 0.5 | 2.7 | 0.989 | 0.989 | 0.989 | 14.4 |
| | 6 | 1256 | 4.7 | 0.5 | 4.6 | 0.987 | 0.983 | 0.984 | |
| WDX12 | 3–5 (test set) | 600 | 2.2 | 0.5 | 8.8 | 0.977 | 0.976 | 0.977 | 15.6 |
| | 6 | 1256 | 4.8 | 0.5 | 10.1 | 0.975 | 0.973 | 0.975 | |

Datasets 1 and 2 denote different RNA templates, respectively barcoded with DPC's four RTA barcodes, whereas datasets 3–6 refer to distinct RNA templates barcoded with optimized WDX RTA barcodes (WDX4: 4 barcodes used; WDX12: 12 barcodes used). Details on the data sets are provided in Supplementary Table 1. Precision and recall reflect averages over barcode classes. Reported run times (CPU-only) refer to 10-fold cross-validated average computing time from 4000 reads on 8 CPU cores (11th Gen Intel(R) Core(TM) i7–1165G7 (2.80 GHz) with 32GB memory).

variation. We demonstrate the benefits of WarpDemuX multiplexing by rapidly phenotyping two genetically distinct SARS-CoV-2 viruses, namely SARS-CoV-2 wild-type (Wuhan-Hu-1) and a variant expressing GFP in ORF8a. We extracted RNA from cells at 8, 24, 48, and 72 hours post-infection (hpi) as well as from viral supernatant at 72 hpi (Fig. 3a and Supplementary Fig. 8). Cells of two uninfected mock samples served as negative controls. Samples were barcoded with WDX12 barcodes and sequenced on a single Minion R9.4.1 flow cell (Fig. 3a; also see Supplementary Note 4). Of the 363,283 reads passing a basecalling quality score (qscore) threshold of 10, we successfully identified barcodes (passing WDX QC, non-noise class prediction) in 95.4% (346,691) of reads (Supplementary Fig. 9a, b). Upon demultiplexing, the acquired reads exhibited a more than 50-fold difference between supernatant samples and some cellular samples (Supplementary Fig. 10). In such highly skewed datasets, even a small fraction of misclassified reads from high-abundance samples could dominate the signal in low-abundance samples. We, therefore, applied a stringent confidence threshold of 0.99, which retained 69% of passing reads (236,952 reads), to minimize cross-sample contamination in the underrepresented supernatant samples (Supplementary Fig. 11) and ensure reliable biological interpretation. At this setting, over 90% of reads from supernatant samples were of SARS-CoV-2 origin (Supplementary Fig. 12). Moreover, EGFP sequences were limited to the SARS2-EGFP-infected samples, while those with WT ORF 8a were almost exclusively present in SARS2-WT-infected samples. Together, these data indicate excellent classification performance, and minimal crossover, even on unbalanced samples representing <1% of the total reads.

We next investigated the read length and read identity distributions of the demultiplexed samples (Fig. 3b–g). We observed clear differences in read length distributions as the infection progressed (Fig. 3b). Discrete bands became visible at 24–72 hpi (Fig. 3b and Supplementary Fig. 13), which could be attributed to distinct subgenomic RNAs by TRS-leader sequence annotation (Supplementary Fig. 14). This analysis showed striking differences in the relative abundances of individual subgenomic RNAs (sgRNAs), with the most abundant sgRNA N present at 22.3% (95% CI = 21.9, 22.7; 48 hpi), compared to the least present subgenomic RNA 10, which was present at $2.5 \times 10^{-1}$% (95% CI = $2.0 \times 10^{-1}$, $3.0 \times 10^{-1}$; 48 hpi) (Fig. 3d). The relative composition of sgRNAs, however, remained relatively stable throughout the infection cycle (Fig. 3d). Importantly, the ability to pool RNA from infected cells on a single flowcell allowed us to assess the replication kinetics of these viruses, without running multiple sequencing experiments. For both viruses, SARS-CoV-2 RNA was first detectable at 24 hpi, and reached a maximum at 48 hpi, comprising

approximately 50% of all poly-adenylated RNA in the RNA extract, before declining to 30% at 72 hpi (Supplementary Figs. 15, 16). No apparent difference in kinetics between the wildtype and EGFP-expressing virus was observed, indicating insertion of EGFP does not effect viral replication over the time frame tested (Fig. 3c).

Finally, we performed a poly(A) tail length analysis, as this is known to regulate RNA translation and stability, and is regulated during infection in other Coronaviruses (Fig. 3e–g). The poly(A) tail length of SARS-CoV-2 sgRNA was previously characterized for a single time point of 24 hpi[14], revealing two tail length populations with a low abundance of shorter tails. Using WarpDemuX, we were able to study the dynamics of poly(A) tail length across multiple time points and compare RNA from cells and virions. This revealed a broad tail length distribution for Vero mRNA, which was stable over time, with a median of 70 nt (2.5th-97.5th percentile: 14–227 nt; 48 hpi), while SARS-CoV-2 RNA had significantly shorter poly(A) tail lengths (median 56 nt, 2.5th–97.5th percentile: 23–108 nt; 48 hpi; two-sided, independent $t$-test, $p < 0.001$), and exhibited regulation of tail length over the course of infection. Intriguingly, at 24 hpi, > 99% of cellular SARS-CoV-2 RNA was attributed to a population with a median tail length of 56 nt, with an additional population with a shorter poly(A) tail length around 29 nt becoming more abundant over time, increasing to 3% at 48 hpi and 15% at 72 hpi (Fig. 3e and Supplementary Fig. 17). These shorter tails in the cell appear to be limited to sgRNAs (Fig. 3f) and were not observed in virion RNA (Fig. 3g). The increased presence of shorter tail lengths at the later time points could reflect mRNA degradation upon cell death, compared to genomic RNA protected within membrane vesicles. In agreement, when compared to cellular RNA at 72 hpi, supernatant RNA within virions had a longer average tail length (52 nt cellular vs. 58 nt supernatant) (Fig. 3g).

## Barcode balancing and enrichment of low abundance samples by adaptive sampling

Adaptive sampling, a unique feature of nanopore sequencing, allows real-time, programmable sequencing decisions (Fig. 4a)[12,15]. This method operates by briefly inverting the voltage across the pore to reject unwanted molecules. One of its main applications is barcode balancing, where rejection aims to unify read counts or total yields between barcodes during sequencing (Fig. 4a)[8]. Most commonly, adaptive sampling decisions are made on basecalled nucleotide sequences[8,15–19]. Although basecalling approaches effectively determine nucleotide sequences, the models used for basecalling require batching to optimize hardware utilization. This batching necessitates the accumulation of a sizable amount of raw signal data before analysis can commence, introducing inherent latency in the decision-

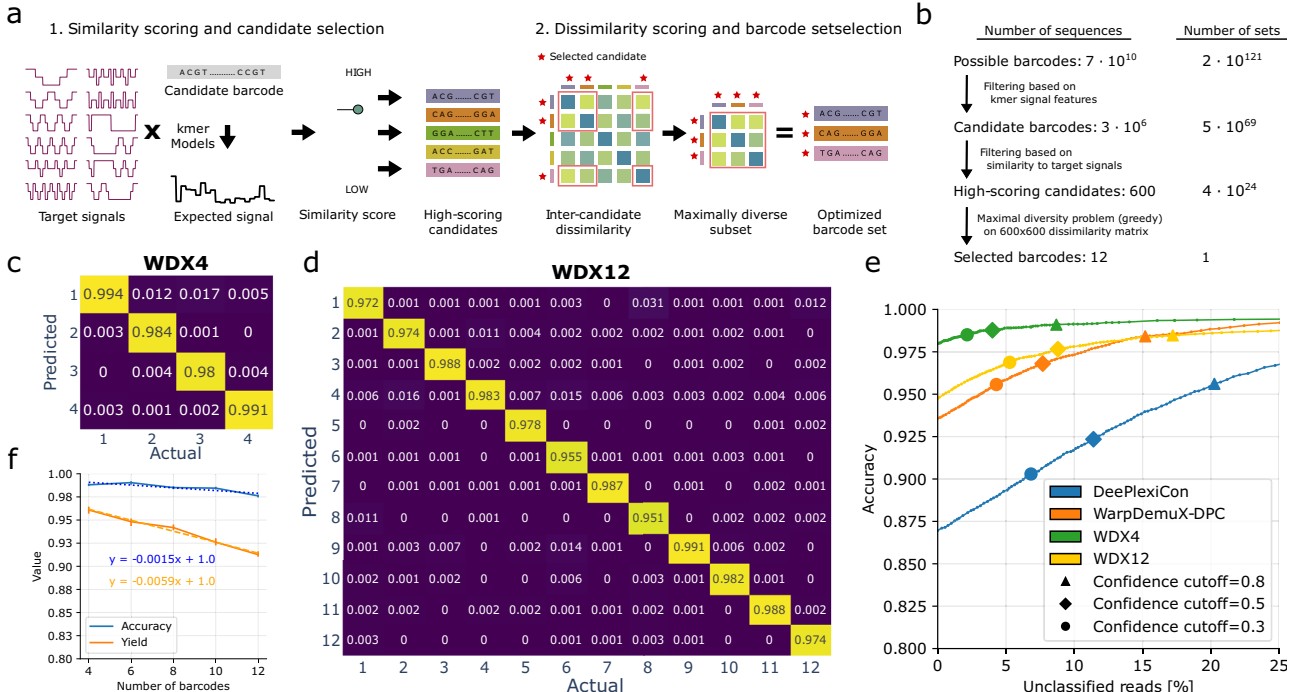

**Fig. 2 | Optimized barcodes improve WarpDemuX performance. a, b** In silico barcode design. **a** Distinct wave-like signal patterns enhance the distinction between the individual target patterns. First, candidate barcodes are scored based on their similarity to a target pattern. Second, high-scoring candidates are selected, and an optimized set of barcodes is identified based on their mutual Dynamic Time Warping Distance (DTWD). **b** Identifying barcode sets using k-mer models. Identification of 12 barcodes with maximal pairwise DTWDs. **c, d** The use of optimized barcodes improves accuracy and yield. Values reported reflect the recall (confusion matrix, normalized over the actual class; true label), on both test sets combined, evaluated at a confidence cutoff of 0.5. **c** 4 optimized barcodes (WDX4) (**d**) 12 optimized barcodes (WDX12). **e** Relationship between accuracy and percentage of unclassified reads, modulated by the confidence cutoff. Values reflect the performance per model as the weighted average over their respective test sets. **f** Performance trends for accuracy and yield with the increasing number of barcodes, evaluated at a confidence cutoff of 0.5. Linear regression analysis of the 10-fold cross-validated performance (weighted averaged over test sets) suggests scalability for a larger set of optimized barcodes. Measured performance is shown by solid lines (mean ± std), and linear fits are represented by dashed lines, with annotated formulas. $R^2$ values are 0.75 (accuracy; blue) and 0.97 (yield; orange). Source data are provided as a Source Data file.

making process. Such delays allow molecules to traverse further through the pore before a potential rejection decision can be made. Extended pore occupancy can impact pore lifetime and sequencing yield, reducing the effectiveness of the approach[20]. Reducing inherent latency in adaptive sampling decision-making can enable more rapid molecule rejection, helping to preserve pore integrity and potentially improve overall enrichment and/or yield of low abundance samples.

Leveraging WarpDemuX's classification speed, we implemented ultra-rapid adaptive sampling for dRNA-seq, aiming to make rejection decisions while the poly(A) tail of RNA molecules is still within the pore. We modified MinKNOW software to provide signal data earlier to the WarpDemuX pipeline, which required adaptations to internal read classification logic (Supplementary Note 5 and Supplementary Figs. 18–20). We also developed our own decision logic to begin rejecting reads once specific thresholds are reached. For this, we use dynamic thresholds that update based on the current sequencing distribution and guide the rejection of reads based on real-time metrics such as adapter counts, read counts, and number of bases (Supplementary Note 6). We applied these decisions to 40 channels each on a MinION R9.4.1 flow cell (Fig. 4b). Notably, 81% of classifications finished within 60 ms after detection of a poly(A) tail, and 98% of classifications completed within 200 ms (Fig. 4c and Supplementary Figs. 21–23). Assuming a translocation speed of 70 nt/s, classification occurs well within 14 nt of the start of the poly(A) tail. In practice, due to the latency of the different softwares, 50% rejections are made within 14 nt and 95% within 34 nt, which would be within the poly(A) tail for the majority of reads (Fig. 4d).

To provide a rigorous test of the adaptive sampling pipeline, we turned to our SARS-CoV-2 samples, which were heavily unbalanced due to differing amounts of RNA present in cells and supernatant (Fig. 4b). All barcode balancing strategies led to a greater number of molecules being sampled, due to higher pore turnover (Fig. 4e, Supplementary Figs. 24–26). Importantly, the rejection of high abundance barcodes not only led to a more even barcode coverage (Supplementary Table 13), reducing the Gini coefficient from 0.44 to 0.15, but also to the enrichment of low abundance barcodes by 33–73% (Fig. 4f and Supplementary Fig. 25).

**Adaptation to RNA004 chemistry maintains high performance**

During the development of WarpDemuX, Oxford Nanopore Technologies released SQK-RNA004, a new direct RNA sequencing chemistry featuring an RNA-specific pore and faster motor protein. While the library preparation approach remains consistent between versions, the new RNA-specific pore exhibits different signal characteristics that necessitate adaptation of our model[21].

To evaluate WarpDemuX's compatibility with the new chemistry, we generated RNA004 datasets using our established dual-barcoding strategy with WDX-RTA barcodes (Supplementary Table 8 and Supplementary Table 14, 15). After retraining our models, we observed robust performance across all barcode configurations. Smaller sets with 4–6 barcodes (WDX4-WDX6) achieved an accuracy of 99.5% with yields between 90–95%. The larger sets (WDX8-WDX10) maintained strong performance at 99% accuracy and 87–91% yield, while WDX12 demonstrated 98–99% accuracy with yields of 90% and 79% (Table 2 and Supplementary Table 16–18).

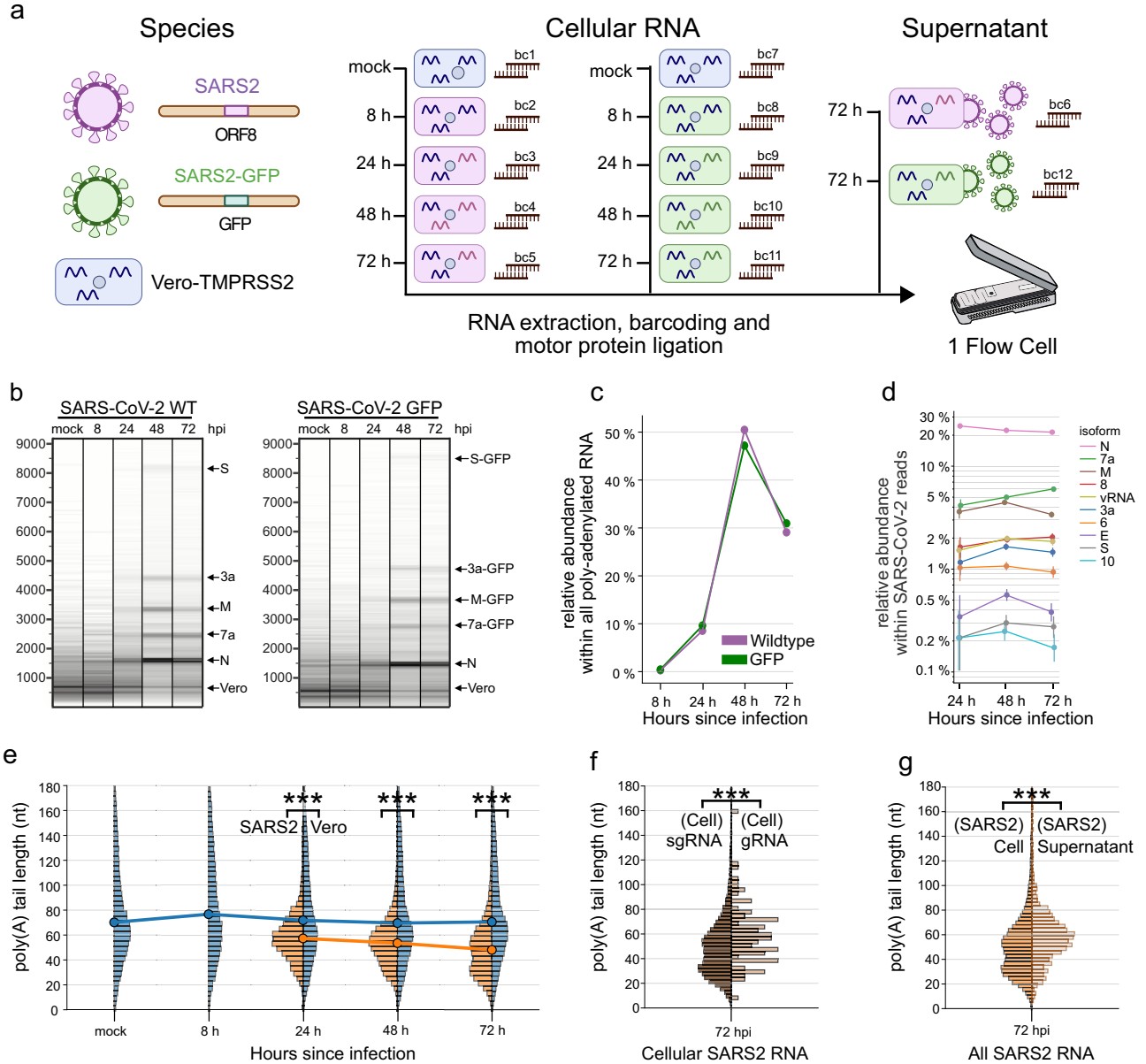

**Fig. 3 | Analysis of SARS-CoV-2 infection kinetics with WarpDemuX.**
**a** Experimental workflow. Cellular RNA was extracted from uninfected Vero-TMPRSS2 cells and Vero cells infected with rSARS-CoV-2 or rSARS-CoV-2-TurboGFP at different time points after infection, as well as from cleared supernatant of infected cells. The extracted RNA was barcoded with distinct WDX12 barcodes and sequenced. Created in BioRender. Smyth, R. (2025) https://BioRender.com/qvl8swr. **b** Virtual gels (read length scaled on a log scale, weighted by number of bases) of the WDX-demultiplexed cellular samples. Distinct bands are labeled according to transcripts matching the size. **c** Relative abundance of SARS-CoV-2 RNA within all reads across time points for the WT and EGFP variants, respectively. **d** Relative abundance of full-length SARS-CoV-2 sgRNA within all SARS-CoV-2 RNA (pooled analysis of wildtype and EGFP variants). Data points reflect the observed

proportion of reads, error bars indicate the 95%CI, calculated using the Clopper-Pearson method. $n = 4651, 40314, 25421$ for 24, 48, and 72 hpi, respectively. **e** Poly(A) tail distribution of cellular RNA per species. SARS-CoV-2 represents pooled samples from WT and GFP variants (***: $p < 0.001$, LTR 8.8e-226, 0.0, 0.0; independent two-sided $t$-test with $n_{Vero} = 46456, 40751, 59569$ and $n_{SARS2} = 4638, 40276, 25359$ for 24, 48 and 72 hpi, respectively). **f** Poly(A) tail distribution of cellular SARS-CoV-2 sgRNA and genomic RNA (gRNA), gRNA defined as having > 8.5 kb read length (***: $p < 0.001$, 1.4e-4; independent two-sided $t$-test, with $n_{sgRNA} = 9614$ and $n_{gRNA} = 87$). **g** Poly(A) tail distribution of cellular vs. supernatant SARS-CoV-2 RNA (***: $p < 0.001$, 3.6e-12; independent two-sided $t$-test, with $n_{Cell} = 25359$ and $n_{Supernatant} = 909$). Source data are provided as a Source Data file.

To provide users with precise control over the accuracy-yield balance, we developed a barcode-specific confidence score calibration system that allows users to select predefined accuracy levels (see Methods). Independent test sets validated this approach, showing close alignment between expected and observed accuracy levels across all barcodes (Supplementary Figs. 27–28). Comprehensive performance metrics, including confusion matrices, precision-recall measurements, and ROC curves, further validate our models' capabilities (Fig. 5a–f and

Supplementary Figs. 29–30). A significant advancement in the RNA004 model was achieved through the improved RNA translocation speed on RNA004, which effectively reduced the length of the signal per adapter to analyze. In addition, the integration of a faster method for adapter detection contributed to these enhancements. Together, these improvements resulted in a remarkable 10-fold increase in processing speed, enabling the demultiplexing of 100,000 reads in just 2.2–3.5 minutes on standard laptop hardware (Fig. 5g).

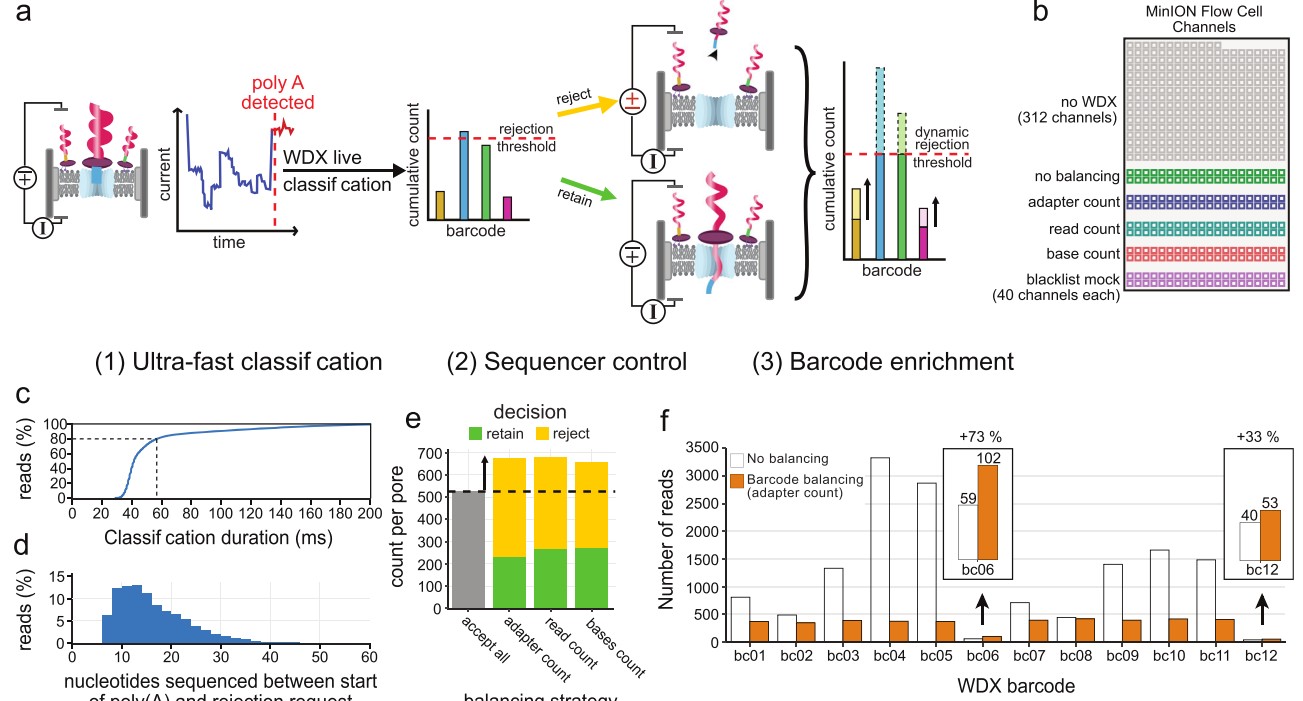

**Fig. 4 | Live barcode balancing using WarpDemuX. a** Principle of live barcode balancing. Raw signal data is accumulated and, once the start of a poly(A) tail is detected, submitted to the WarpDemuX live classifier. Based on its classification, a decision is taken to either retain or reject the barcode, resulting in increased read turnover when rejecting. **b** Channel assignments for live barcode balancing during the experiment, with 40 channels per WarpDemuX balancing strategy for comparative analyses. The no-WDX channels served as internal control. **c** Cumulative distribution plot of duration of WarpDemuX barcode classification itself.

**d** Histogram for number of nucleotides that passed through the pore from the start of poly(A) until the rejection decision. **e** Bar plot of the number of adapter counts per pore per balancer, colored by the decision taken after classification of the adapter. (**f**) Bar plot showing the total number of reads recovered with- and without barcode balancing (40 channels each). Inset shows enrichment for low abundance samples. Here we show the results for the adapter count balancing strategy, while results for the other strategies are depicted in Supplementary Fig. 25. Source data are provided as a Source Data file.

## Discussion

Despite its transformative potential for RNA biology research, dRNA-seq has been limited in its applications and wider adoption by the lack of effective multiplexing techniques. To address this, we developed WarpDemuX, a signal-based barcode-design and demultiplexing method which exhibits high classification speed, accuracy, yield, and scalability. By leveraging these characteristics, we used WarpDemux to implement adaptive sampling for dRNA-seq, which enables barcode balancing and the enrichment of low abundance samples.

WarpDemuX introduces two key innovations: (i) a signal pre-processing step and (ii) a lightweight SVM-DTWD classifier. Signal pre-processing reduces data noise and mitigates spurious temporal correlations from varying dwell times (Supplementary Note 7 and Supplementary Fig. 31). The SVM-DTWD classifier requires minimal training data (we used 400 reads per barcode, see Supplementary Note 9 and Supplementary Figs. 32–33), and is computationally inexpensive. While we validated WarpDemuX using combinations of inline RNA barcodes and transcript types as ground truth labels, the method is flexible and can be trained on any set of transcripts serving as ground truth. Related, signal-based de-multiplexing tools exist: For example, HycDemux denotes a demultiplexing tool for nanopore-based DNA sequencing that also employs DTW of raw signals. However, it cannot be employed in real-time settings due to its cluster-based demultiplexing procedure[22]. DPC, the only tool for RNA002 dRNA-multiplexing, converts raw signals into images for classification with a deep convolutional neural network[23] and was less accurate, as well as 10 times slower than WarpDemuX. In all these approaches, DNA barcodes embedded in the standard (DNA-based) sequencing adapter are used to demultiplex RNA sequences. The RNA basecaller's inability to process DNA barcodes limits direct comparisons between signal-level

analysis and basecalling-based methods. Should ONT introduce RNA-based barcodes in their sequencing protocols, for example, through the use of RNA-based sequencing adapters, it would enable comprehensive benchmarking studies comparing signal-based and basecalling approaches, while also allowing investigation of potential library preparation biases between the use of DNA and RNA barcodes. Additionally, such studies could investigate the risk of overfitting in basecalling models, given the relatively small sequence space of barcodes compared to full-length RNA sequences. Whilst this manuscript was in review, Pryszcz et al. developed a custom, GPU-accelerated RTA basecaller (SeqTagger) for post-acquisition demultiplexing, that infers the specific nucleotide sequences of DNA barcodes placed in the RTA adapter[24]. For RNA004, the authors provide a 4-barcode model, which achieves comparable accuracy to WDX4, but processes reads five times slower at 10,000 reads/minute, versus WDX4's 50,000 reads/minute (Fig. 5g) while using an NVIDIA RTX A4000 GPU versus an Intel i7–1165 G laptop CPU for WarpDemuX, thus requiring substantially more compute resources.

WarpDemuX's ultra-fast classification allowed us to implement barcode balancing for the uniform sequencing of multiplexed, uneven samples. Since we were able to make read decisions < 200 ms after the start of the poly(A) tail, virtually all poly-adenylated RNA were rejected before sequencing the main body of the RNA (also see Supplementary Note 9, Supplementary Fig. 34 and Supplementary Table 19). Our fast decision time may have prevented pore blockage because early rejection preempted the formation of RNA structures on the trans side of the pore, which may otherwise hinder the ejection of the molecule back through the pore. In addition to the enrichment of low abundance barcodes through higher pore turnover, dynamic barcode balancing could also be used to extend flow cell lifespan and total yield by

**Table 2 | Performance of WarpDemuX on SQK-RNA004 for two distinct test sets**

| Model | Runtime (ms/read) | Dataset | Noise reads (%) | Target accuracy — Observed accuracy | | | | | | | Target accuracy — Unclassified reads (%) | | | | | | |
|---|---|---|---|---|---|---|---|---|---|---|---|---|---|---|---|---|---|
| | | | | 0.95 | 0.96 | 0.97 | 0.98 | 0.99 | 0.995 | 0.999 | 0.95 | 0.96 | 0.97 | 0.98 | 0.99 | 0.995 | 0.999 |
| WDX4 | 1.3 | Test set 1 | 1.7 | 0.98 | 0.98 | 0.98 | 0.98 | 0.99 | 0.99 | 1.00 | 0.1 | 0.1 | 0.1 | 0.7 | 2.6 | 5.7 | 29.7 |
| | | Test set 2 | 1.8 | 0.98 | 0.98 | 0.98 | 0.98 | 0.99 | 0.99 | 1.00 | 0.1 | 0.1 | 0.1 | 0.7 | 2.6 | 5.5 | 30.0 |
| WDX6 | 1.5 | Test set 1 | 1.7 | 0.97 | 0.97 | 0.97 | 0.98 | 0.99 | 0.99 | 1.00 | 0.1 | 0.3 | 0.8 | 2.5 | 6.0 | 10.6 | 38.6 |
| | | Test set 2 | 1.6 | 0.97 | 0.97 | 0.97 | 0.98 | 0.99 | 0.99 | 1.00 | 0.1 | 0.4 | 0.8 | 2.5 | 6.0 | 10.7 | 38.5 |
| WDX8 | 1.7 | Test set 1 | 1.9 | 0.96 | 0.96 | 0.97 | 0.98 | 0.99 | 0.99 | 1.00 | 0.2 | 0.7 | 1.5 | 4.0 | 9.3 | 21.8 | 51.4 |
| | | Test set 2 | 1.9 | 0.96 | 0.96 | 0.97 | 0.98 | 0.99 | 0.99 | 1.00 | 0.2 | 0.7 | 1.6 | 4.1 | 9.6 | 22.2 | 51.7 |
| WDX10 | 1.9 | Test set 1 | 2.0 | 0.96 | 0.96 | 0.97 | 0.98 | 0.99 | 0.99 | 1.00 | 0.7 | 1.5 | 3.6 | 6.3 | 12.5 | 22.8 | 58.3 |
| | | Test set 2 | 2.0 | 0.95 | 0.96 | 0.97 | 0.98 | 0.99 | 0.99 | 1.00 | 0.6 | 1.5 | 3.6 | 6.6 | 13.0 | 23.4 | 58.8 |
| WDX12 | 2.2 | Test set 1 | 2.1 | 0.95 | 0.96 | 0.97 | 0.98 | 0.99 | 0.99 | 0.99 | 2.0 | 3.4 | 5.8 | 10.2 | 20.8 | 35.8 | 63.2 |
| | | Test set 2 | 2.1 | 0.95 | 0.96 | 0.97 | 0.98 | 0.99 | 0.99 | 1.00 | 2.1 | 3.5 | 6.0 | 10.5 | 21.4 | 36.2 | 63.7 |

Test set 1 incorporated data from experimental runs 7–9, while Test set 2 consisted of runs 10 and 11. Details on the data sets are described in Supplementary Table 14. Barcodes per model are described in Supplementary Table 16. The reported metrics represent averages across barcode classes. All benchmarks were performed with an 11th Gen Intel (R) Core (TM) i7-1165G7 processor (2.80 GHz, 32GB RAM) using 8 CPU cores.

dynamically rejecting reads from barcodes that generate high numbers of blocked pores (Supplementary Note 5 and Supplementary Note 6). Implementations for demultiplexing and adaptive sampling are available at Github [https://github.com/KleistLab/WarpDemuX].

WarpDemuX showed high classification accuracy, even on barcodes that were not explicitly optimized for DTWD-based separability. Further improvements in performance were achieved by designing 12 barcodes that were maximally separated in signal space, which can likely scale to at least 24 barcodes whilst maintaining a classification accuracy above 95%. Our design strategy is not inherently limited by the number of barcodes, however, increasing the total number of barcodes impacts classification confidence and yield. We, therefore, acknowledge the possibility of reaching a saturation point in creating distinct signal patterns, which would set an upper limit on the number of employable barcodes. Lengthening the barcodes could effectively push back this saturation point, allowing for the implementation of substantially larger barcode sets before reaching capacity limitations while enhancing barcode classification confidence for smaller sets. Systematic efforts to test the limits of barcode design and classification could include the development of signal (k-mer) models for DNA sequenced in the 3′ to 5′ direction, as this would allow more accurate in silico design of distinct barcodes. Furthermore, as the number of DTWD calculations for classification scales with the number of barcode classes, the use of learned DTW kernels rather than full signal-to-signal comparisons could be explored to further speed up the method.

An important feature of our approach is that it operates independently of basecalling, and is thus not influenced by continued development of new basecallers. However, changes in sequencing chemistry will necessitate retraining of the SVM-DTWD kernel. On this note, ONT unveiled the new direct RNA sequencing kit SQK-RNA004 in 2024. This newer chemistry contains an RNA-sequencing-specific pore with faster motor protein. Notably, the library preparation approach remained unaffected for the new sequencing kit, allowing seamless integration of our DNA barcodes. However, the new RNA-specific pore displays different signal characteristics. We therefore adapted WarpDemuX for RNA004, achieving high performance across different barcode set sizes (99.5% accuracy with 90–95% yield for WDX4-WDX6; 99% accuracy with 87–91% yield for WDX8-WDX10; and 98–99% accuracy with 79–90% yield for WDX12). Notably, these performances were achieved using barcodes optimized for the previous (RNA002) sequencing chemistry, suggesting that further improvements of WarpDemuX could be achieved through RNA004-specific barcode optimization. By integrating ADAPTed's (our custom tool for DNA adapter signal detection; https://github.com/KleistLab/ADAPTed) learned convolutional classifier for RNA004-specific adapter detection, we also improved processing speed to 2.2 (WDX4) to 3.5 (WDX12) minutes per 100,000 reads on a standard laptop (8 cores, run time scales linearly with cores). Furthermore, we implemented barcode-specific confidence score calibration, allowing users to precisely control accuracy-yield trade-offs according to their experimental needs.

We demonstrated the application of dRNA-seq barcode-specific adaptive sampling and barcode balancing on SARS-CoV-2 infected cells. This enabled us to measure the replication kinetics of two different SARS-CoV-2 viruses by direct readout of viral RNA in a single rapid experiment. Standard replication assays using plaque assay take several additional days after sample collection[25,26], meaning that WarpDemuX can aid in rapidly assessing the replication capacity of novel SARS-CoV-2 variants as they emerge[27,28]. We were also able to measure the relative abundance RNA isoforms and poly(A) tail length of SARS-CoV-2 samples taken at different time points. Altered subgenomic RNA expression patterns have been associated with SARS-CoV-2 lineages[28,29], infection states[30], and can be regulated by host proteins[31]. WarpDemuX would enable this phenomenon to be investigated without sequencing batch effects. Similarly, changes in poly(A)

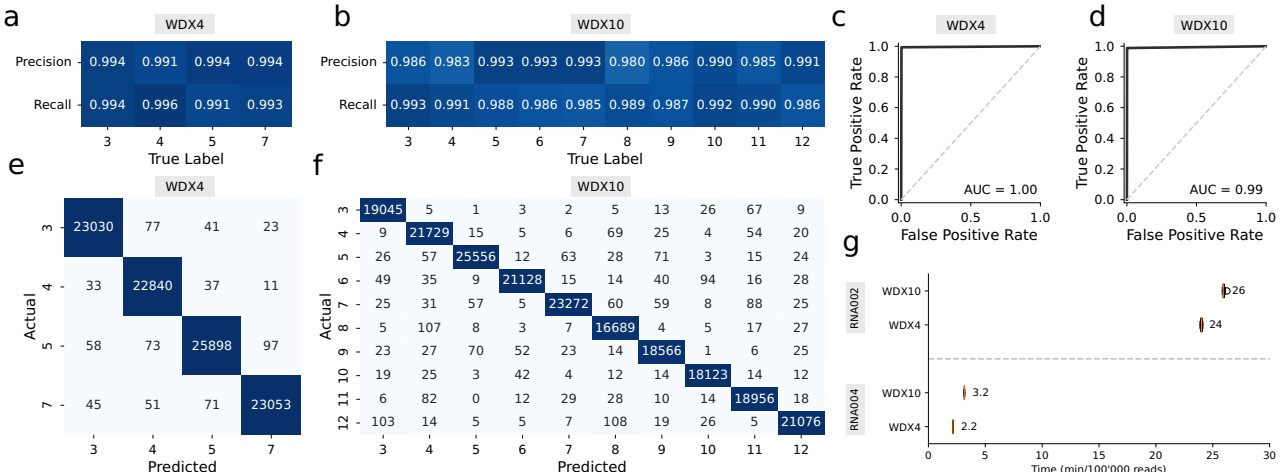

**Fig. 5 | Performance evaluation of WarpDemuX on SQK-RNA004 for WDX4 (99.5% target accuracy, 95% yield) and WDX10 (99% target accuracy, 87% yield) models on Test Set 1 and 2 combined.** Details on the data sets are provided in Supplementary Table 14. Barcodes used per model are described in Supplementary Table 16. **a, b** Precision and recall metrics for individual barcodes in WDX4 and WDX10 configurations, respectively. **c, d** Receiver operating characteristic (ROC) curves for WDX4 and WDX10 configurations (black lines) vs. chance (diagonal gray dashed line). **e, f** Confusion matrices showing classification accuracy and cross-talk between barcodes for WDX4 and WDX10 sets. (**g**) Processing speed benchmarks on 8 CPU cores (11th Gen Intel(R) Core(TM) i7–1165G7, 2.80 GHz, 32GB RAM). Distribution of processing time per read (in min/100'000 reads) for 10 runs. Median shown in orange, the box indicates the interquartile range (IQR), and whiskers denote 1.5 × IQR with points outside the whiskers (flier points) shown. Source data are provided as a Source Data file.

tail length during infection have been previously observed for other coronaviruses[32,33] and could point to the regulation of poly(A) tail length to affect viral RNA translations or genome packaging.

Important to note is that ONT is not providing a Flongle flow cell for the new RNA004 chemistry, as such, every single sample needs to be sequenced on a (larger) MinION or PromethION flow cell, with multiple rounds of washing, substantially increasing hands-on-time and cost per sample. Our multiplexing strategy reduces the cost per sample below single Flongle flow cells per sample, while increasing the output per sample (Supplementary Note 10 and Supplementary Table 20–23).

In summary, we envision that ultra-fast and high-accuracy dRNA-seq multiplexing will allow for more cost- and resource-effective profiling of RNA, especially when combined with adaptive sampling. As dRNA-seq continues to evolve, tools like WarpDemuX will play a pivotal role in maximizing the benefits of this transformative technology, catalyzing advancements in transcriptomics and epitranscriptomics research.

## Methods
### Adapter signal processing and alignment
To account for the intrinsic variation in the dwell time, as well as the within-event measurement noise, (Supplementary Fig. 35a, b), we partition the raw signal of the adapter into $n$ segments, based on the method introduced in ONT's signal analysis tool Tombo (github.com/nanoporetech/tombo; algorithmic details are included in Supplementary Method 2). Each segment is subsequently represented by its average signal. To mitigate the risk of data loss due to missed events (Supplementary Fig. 35c), we intentionally over-segment the signal by detecting approximately 10% more segments than expected based on the number of bases in the RTA. We normalize the segmented adapter signal and retain only the last 25 segments, which should contain the barcode signal given its known sequence length and relative location within the RTA.

To correct for the likely imperfect segmentation, we perform subsequent Dynamic Time Warping (DTW) to compute the distance between segmented barcode signals. DTW is a robust solution to align two sequences with temporal variation by compressing or stretching their temporal axis to minimize the cumulative Euclidean distance

between corresponding elements (thus correcting for imperfect segmentation).

To compute the Dynamic Time Warping Distance (DTWD) between two segmented signals $s_1$ and $s_2$, $DDTW(s_1, s_2)$ we use the dtaidistance package (2.3.11)[34]. The parameters used are detailed in Supplementary Method 3.

### Barcode design via dynamic time warping distances
We developed a method to generate sets of $k$ RTA barcodes with distinct signal profiles de novo, as it would be computational intractable to perform an exhaustive search over all possible barcode sets of size $k$ (complexity $\binom{4^L}{k}$) for barcodes of length $L$, e.g., $\approx 2.3 \times 10^{121}$ for $L = 18$ and $k = 12$). First, we constructed a set of wave-like target signal patterns that exhibited high inter-target DTWD, since we expect such patterns to improve signal segmentation and barcode separability. Then, we searched for barcode sequences whose putative signal, constructed using the ONT k-mer models (github.com/nanoporetech/kmer_models), would closely match these target patterns. Because there are no DNA k-mer models for DNA sequenced in 3' to 5' direction, we filter for k-mers we expect have similar signal profiles in both sequencing directions, as detailed in Supplementary Method 1. For the twelve target patterns highlighted in Fig. 2a, we selected the top 50 barcodes per target pattern, yielding a total of 600 barcodes, for which we computed all 600 × 600 pairwise DTWDs. Lastly, within the matrix of $600^2$ pairwise DTWDs, we solved the Maximal Diversity Problem (MDP, below) with $k = 12$ to propose 12 candidate barcodes that we expect to be accurately identifiable and distinguishable. The MDP is defined as:

$$\max_{\delta(\cdot)} \quad \sum_{i=1}^{M-1} \sum_{j=i+1}^{M} DDTW(s_i, s_j)\delta(i)\delta(j)$$
$$\text{Subject to} \quad \sum_{i=1}^{M} \delta(i) = k \tag{1}$$
$$\delta(i) = \{0,1\}, \ 1 \le i \le M$$

where $M$ denotes the entire set of barcode candidates (600 in our example), $k$ is the number of distinct barcodes we aim to select, $s_i$ is the segmented signal of barcode $i$, $DDTW(s_i, s_j)$ is the DTWD between $s_i$ and $s_j$, and $\delta(i)$ is a dirac delta function indicating whether barcode $i$ is

selected. Detailed descriptions of the steps in the design process are included in Supplementary Method 1.

### Design, synthesis and sequencing of dual-barcoded RNA for classifier training

To evaluate our barcode design and train a classifier for demultiplexing, we generated labeled RTA signals by synthesizing and sequencing various custom RTA adapters (Supplementary Table 24) attached to RNA molecules with integrated (in-line) RNA barcodes—serving as ground truth labels. Notably, the in-line RNA barcodes used were integrated into the RNA transcripts using PCR, positioning them at the 5' end of the poly(A) tail to enhance the accuracy of basecalling and, consequently, the precision of barcode classification, compared to standard RNA barcodes that are ligated to the 3' end of the poly(A) tail. We randomized the RNA in-line barcode to RTA barcode assignment across replicates to circumvent any potential of bias introduced by the in-line RNA barcode signal. The generated data sets (1–6) are detailed in Supplementary Table 1.

The barcoded RNA designated for training the WarpDemuX classifier was synthesized via in vitro transcription. To enrich the diversity of our dataset, we used a variety of RNA templates. Briefly, starting from different DNA templates, 12 PCRs per template were performed with a forward primer containing a T7 RNA polymerase promoter and varying reverse primers to add different RNA barcodes to the 3' end, followed by a 10nt poly(A) tail (Supplementary Fig. 36 and Supplementary Table 25, 26). After purification of PCR products, they were used as templates for in vitro transcription, followed by DNA digest and RNA clean-up. The RNA was then further polyadenylated to ensure no signal spillover from the RNA barcode into the DNA signal. Further details and a visual summary of the protocol are included in Supplementary Method 4.

The direct RNA sequencing library preparation was started by ligating 100 ng of poly-adenylated barcoded RNA onto a custom RTA adapter. Further library preparation was performed according to ONT's recommendations, with the exception of the use of MarathonRT as a reverse transcription enzyme (see Supplementary Method 5). After motor protein ligation, 7 μL of the prepared library was loaded on a Flongle FLO-FLG001 flow cell, and reads were acquired until all pores became inactive. For a detailed description of all steps, see Supplementary Method 5. Reads were basecalled post-acquisition with the direct RNA model of ONTs research basecaller Dorado. See Supplementary Method 6 for the full configuration.

### WDX-RTA DNA barcode identification

We trained a support vector machine (SVM) to classify the sequenced dual-barcoded RNA reads into RTA barcode categories based on their RTA-barcode signal, using a combination of read mapping and RNA barcode classification to infer their ground truth labels.

To determine ground truth labels, reads were first uniquely mapped to RNA reference sequences with LAST[35]. Then, the RNA barcode region was identified and reads were associated with the RNA barcode that had the smallest Levenshtein distance to them. Reads for which this distance surpassed a defined threshold of 4 were discarded. Based on the specific combinations of RNA and RNA-RTA barcode pairs used during library preparation, the read assignments were used to establish the RTA-barcode ground truth labels required to train the classifier.

### Data set selection

Datasets 1 and 2 served as the test sets for the WarpDemuX-DPC and DPC models. For the WarpDemuX models, we designed two distinct test sets: one comprising 600 reads per barcode from datasets 3–5, and another using the entirety of dataset 6, which contains RNA without enzymatic poly-adenylation.

We constructed validation sets for our models, comprising 200 reads per barcode, utilizing data from replicate 2–5 of the DeePlexiCon training data, as detailed in Smith et al.[7], for the WarpDemuX-DPC model, and data from datasets 3–5 for the WarpDemuX models (WDX4, WDX12).

Training sets, consisting of 400 reads per barcode, were sampled from data from replicate 2–5 of the DeePlexiCon training data[7] for the WarpDemuX-DPC model, and our datasets 3–5 for the WarpDemuX models (WDX4, WDX12). First, 400 training instances for the noise class were selected based on the robust Z-score (defined below) of the median inter-class distance per read. Identification of noise instances is further detailed in Supplementary Note 1. Then, to ensure the training set captured the full range of the original data distribution despite its limited size, we used greedy approximations of the Maximal- and Minimal Diversity Problem, stratified across barcode classes (implementation is included in Supplementary Note 1), to select $2 \times 80$ instances per barcode that respectively maximized and minimized their mutual distance. The training set was completed with 240 randomly sampled instances per barcode. Supplementary Materials provide further details on training set size determination (Supplementary Note 1).

The robust Z-score is calculated as follows, where $X_i$ represents the value under evaluation (median inter-class distance), Median($X$) is the dataset's median, and MAD is the Median Absolute Deviation:

$$\text{Robust Zscore} = \frac{X_i - \text{Median}(X)}{\text{MAD}} \quad (2)$$

$$\text{MAD} = \text{Median}\left(\left|X_i - \text{Median}(X)\right|\right) \quad (3)$$

All sampling processes for validation, training, and test sets were stratified across replicate groups to compensate for variations in sequencing yields, ensuring a balanced representation across all data.

### Barcode classification

The RTA barcode classifiers were trained on preprocessed RTA barcode signals. To obtain these RTA barcode signals, we first identify the DNA adapter signal in the raw signal using ADAPTed (our custom tool for DNA adapter signal detection; https://github.com/KleistLab/ADAPTed; LLR-based method for RNA002, convolutional classifier method for RNA004). Adapter detection is necessary but also functions as a signal quality filter: ADAPTed processing entails poly(A) detection and various signal characteristics checks. After successfully identifying the DNA adapter signal, we segment, normalize, and trim it, as described in the "Adapter signal processing and alignment" section. For each RTA barcode signal, we generated a feature representation that quantifies how similar the signal is to the RTA barcode signals of reads with known labels. To do this, DTWDs between the signal and all labeled signals in the training data set are computed.

We then use Support Vector Machines (SVMs) for classification by introducing a Dynamic Time Warping Kernel function, $K$, which is defined as:

$$K(s_1, s_2) = \exp\left(-\gamma \cdot DDTW(s_1, s_2)\right) \quad (4)$$

where $s_1$ and $s_2$ are two preprocessed RTA barcode signal instances, and $\gamma = 1.2$ is a hyper-parameter of the kernel function.

During training, the SVM kernel matrix is based on the pairwise DTWDs between the barcode signals in the training data.

We use an SVM implementation based on libsvm[36], with a one-vs-one scheme for multi-class support, as implemented in Scikit-Learn (1.3.1)[37]. Herein, calibrated class membership probability estimates are provided based on an internal 5-fold cross-validation combined with a multiclass extension to Platt scaling[38]. See Supplementary Method 3 for further details on the model parameterization.

During inference, for each read to be classified, $K(s_1, s_2)$ is computed against all barcode signals in the training data, i.e., $s_2 \in B$, with $B = \{B_1, B_2, \ldots B_k\}$, where $B_i$ contains all segmented barcode signals that belong to the RTA barcode $i$ based on their ground truth label.

Each prediction in our classification system is paired with a confidence score, indicating the reliability of the predicted class label. Reads for which the prediction has a confidence score below a user-defined cutoff are discarded (unclassified). For comparability, we adopt the same method as used in DeePlexiCon and define the confidence score as the interval between the classifier's probability estimate of the predicted class and the second most probable class.

## Sequencing of SARS-CoV-2 infected cells

To prepare the sequencing samples, $5 \times 10^5$ Vero E6 TMPRSS2 cells (CVCL_YQ49, a generous gift from S. Pöhlmann; authenticated by the provider) were infected in duplicate with two different recombinant SARS-CoV-2 viruses at MOI = 0.01. Briefly, rSARS-CoV-2 WT and rSARS-CoV-2 GFP inoculum were prepared in DMEM supplemented with 1% FCS[39]. Before the infection, Vero E6 TMPRSS2 cells were washed once with PBS and incubated with the respective inoculum for 1 h at 37 °C with gently shake every 10 min. The inoculum was removed, and fresh DMEM supplemented with 5% FCS, 100 mg/mL of penicillin, and 100 μ/mL of streptomycin was added to the cells.

Then, cells were harvested at 8 and 24 h post infection. A mock control of uninfected cells was also performed. Total RNA was extracted with Trizol (Invitrogen) using the manufacturer's recommendations, precipitated by the addition of 0.1 volumes of 3 M NaOAc and 3 volumes of ice-cold 100 % EtOH and incubation at −20 °C for >12 h, followed by centrifugation at $16{,}000 \times g$ for 30 min at 4 °C, a wash with 70 % EtOH and another round of centrifugation before resuspension in 20 μL RNase-free H2O. RNA concentration was quantifed on Nanodrop, and 1 ug of total RNA per sample was used for WarpDemuX-multiplexed direct RNA library preparation as described in Supplementary Method 5 and Supplementary Table 27, with the exception of the usage of the 100 ms chunk size protocol (see Supplementary Note 5). The generated reads were demultiplexed with the WDX 12-barcode model (WDX12) at a confidence threshold of 0.8, and reads were aligned with minimap (see Supplementary Note 4 for details).

## Barcode-specific adaptive sampling

We used the Read Until API (see https://github.com/nanoporetech/read_until_api) to communicate with the MinKNOW server during sequencing. During sequencing, read data is streamed to Read Until channel caches in fixed-duration chunks, and continually added to as sequencing of the read progresses. WarpDemuX analyzes these caches to detect RTA signals and classify barcodes.

Live demultiplexing begins by identifying the transition from adapter to poly(A) tail in the raw signal, utilizing ADAPTed's fast method tailored for streaming data, after which the read is processed and classified.

To minimize the time spent sequencing unwanted reads, we developed a custom MinKNOW protocol with a reduced read chunk duration (100 ms; approx. 300 observations per chunk). Details are included in Supplementary Note 6. Instructions to setting up MinKNOW for adaptive sampling using our custom protocol are available at Github [https://github.com/KleistLab/WarpDemuX]. Details on the system specifications used for barcode-specific adaptive sampling are included in Supplementary Method 7.

Due to the nature of dRNA-seq sample preparation, not all adapter molecules are ligated to a target molecule, leading to adapter-only reads. These reads, while largely filtered by MinKNOW after sequencing, can skew the real-time WarpDemuX-classified barcode distribution, necessitating correction based on actual sequencer output (pod5

files). WarpDemuX tracks these outputs to maintain an accurate sample distribution and estimate the sequenced base count per sample. The user can determine the aspect to be balanced and the degree of sensitivity applied.

## RNA004 dataset generation and selection

For RNA004 chemistry evaluation, we generated training, validation, calibration, and test datasets from five independent sequencing replicates that were synthesized using our dual-barcoding approach as described in Supplementary Method 5, with the exception of the usage of the RNA004 motor protein RLA, and a PromethION RNA004 flow cell (FLO-PRO004RA). From replicates 1–3, we randomly selected 2000 reads per barcode (stratified across replicates) for initial data distribution analysis and noise class identification. The training set comprised 400 reads per barcode from this set, with noise class instances selected based on robust Z-scores of median inter-class distances as before. To ensure a comprehensive representation of signal variations in the training set, we employed a stratified sampling strategy combining maximally diverse (80 reads), minimally diverse (80 reads), and random (240 reads) instances per barcode.

Validation sets (200 reads per barcode, replicates 1–3) were used to select optimal models for different barcode set sizes (WDX4-WDX12). Two independent test sets were created: test set 1 containing 10,000 reads per barcode from replicates 1–3, and test set 2 combining replicates 4–5 with 10,000–18,000 reads per barcode. A separate calibration set of 2000 reads per barcode from replicates 1–3 enabled confidence score mapping refinement.

## RNA004 confidence score calibration

To establish reliable accuracy estimates, we developed a barcode-specific calibration procedure that maps model confidence scores to predefined target accuracy levels (95–99.9%). For each barcode, we first calculated the baseline error rate ($E_b$) on the calibration dataset. We then determined confidence score thresholds ($T$) that would achieve desired target accuracy levels ($A_t$) by computing the confidence score quantile ($q$) from misclassified reads:

$$q = 1 - (100 - A_t)/E_b \qquad (5)$$

Confidence scores above $T$ were retained to achieve the target accuracy level. This calibration was performed independently for each barcode to account for barcode-specific characteristics and error patterns.

## Reporting summary

Further information on research design is available in the Nature Portfolio Reporting Summary linked to this article.

## Data availability

Raw and basecalled sequencing data are deposited at the European Nucleotide Archive (ENA), under accession PRJEB84366. Source data are provided in this paper.

## Code availability

WarpDemuX was implemented as a Python (3.8) command line tool with Cython-based signal processing to achieve fast CPU run time. Detailed usage and installation instructions, as well as test data, are provided in the code repository. The source code is available at GitHub [https://github.com/KleistLab/WarpDemuX], and a frozen version used in this manuscript (WarpDemuX: v0.4.5) is available via Zenodo[41].

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

## Acknowledgements

The authors would like to thank Anke Sparmann for critical feedback, and the HPC Service of ZEDAT[40], Freie Universität Berlin, and Brouwer Art & Technology for computing time. W.L.-W. was funded by European Union's Horizon 2020 research and innovation program, under the Marie Skłodowska-Curie Actions Innovative Training Networks grant agreement no. 955974 (VIROINF). The research was supported by the Center for Structural Biology of HIV-1 RNA U54 AI170660 and Helmholtz Association VH-NG-1347 to RPS. RPS also acknowledges the interdisciplinary Thematic Institute IMCBio +, as part of the ITI 2021–2028 program of the University of Strasbourg, CNRS and Inserm, IdEx Unistra (ANR-10-IDEX-0002), SFRI-STRAT'US (ANR 20-SFRI-0012), and EUR IMCBio (ANR-17-EURE-0023) under the framework of the French Investments of the France 2030 Program. M.v.K. acknowledges funding by the Deutsche Forschungsgemeinschaft (DFG, German Research Foundation) under Germany's Excellence Strategy—The Berlin Mathematics Research Center MATH + (EXC-2046/1, project ID: 390685689), as well as the German Ministry for Science and Education (BMBF), grant number 01KI2016.

## Author contributions

P. B., M. O.-N., and A.-S.G.-B. performed the wet lab experiments. W. vdT. and P. B. performed the bioinformatic analysis. P. B. conceived the work. R. P. S. and M. vK. supervised the work. R. P. S. and M. vK. acquired funding to conduct the work. W. vdT. and P. B. prepared the figures. W.L.-W. provided substantial feedback throughout the development of the work, including figure preparation and paper writing. W. vdT. and P. B. wrote the paper, with contributions from all authors.

## Funding

## Competing interests

An international priority patent application was filed jointly by Robert-Koch Institute, Helmholtz Zentrum für Infektionsforschung, and Freie Universität Berlin on April 26, 2024, at the European Patent Office (EPO) under number PCT/EP2024/061629, on which W. vdT., P.B., W.L.-W., R.P.S. and M.vK are listed as inventor. The application covers methods for barcode-based molecule identification using dynamic time-warping alignment of electrical signals, as well as the computational approach for designing optimal barcode sets based on signal characteristics. Other authors claim no competing interests.
