## [Peer Review file · Nature Communications]

Demultiplexing and barcode-specific adaptive sampling for nanopore direct RNA sequencing

Corresponding Author: Professor Max von Kleist

Version 0:

Reviewer comments:

Reviewer #1

(Remarks to the Author)

The paper from van der Toorn et al presents a lovely demonstration of the use of DTW for detection of DNA barcode for use in a Nanopore direct RNA library. The manuscript is well written, clear and provides some very useful tools for the community. I only have minor comments on the manuscript (below) aside from one broad comment that may not be feasible for the authors to address (but would be nice if they could!).

The training method used by the authors including the integration of an RNA barcode to compare with the DNA signal based barcode is very nice. The only experiment I'd like to see in an ideal world is a "barnyard" type experiment where barcodes were used on RNA derived from different species. This would allow the species to act as the control barcode and thus the possibility of specific sequences interfering with the barcode detection could be tested. However - this is not an absolute requirement - but would be nice to see!

1. The authors cite Kovaka et al as evidence that signal based approaches are faster for read classification than base calling approaches. This claim is not supported by the work of Kovaka et al when considered in context. The paper from Payne et al that was published back-to-back with Kovaka et al demonstrates that base calling outcompetes signal based approaches in these contexts. I was going to ask why the authors had not compared their methods with direct basecalling approaches but then realised the elegance of the approach of using DNA barcodes in an RNA sequencing context. Therefore I suggest that the authors simply de-emphasize these comments.

2. The authors should clarify that their method cannot be benchmarked against base calling due to the mixed model of DNA and RNA. It should also be clarified that if RNA barcodes were made available by ONT then this experiment would be possible.

3. If I have followed the paper correctly, the warp demux approach only allows classification based upon barcode and not based upon a specific molecule within the sample. If this is not the case, it would be nice to clearly see that shown in the manuscript.

(Remarks on code availability)

The code is well written, well commented and documented.

I have not run the code.

Reviewer #2

(Remarks to the Author)

In this manuscript, van der Toorn et al. developed a novel method for designing and detecting RTA barcodes in nanopore direct RNA sequencing, addressing limitations in current multiplexing techniques. Their new design strategy optimizes 12 RTA barcodes to maximize inter-barcode differences in signal space. The barcode signal is analyzed by signal segmentation using the Tombo strategy, distance matrix calculation via DTW, and final classification using SVM. WarpDemuX enables barcode-specific adaptive sampling and balancing for real-time, programmable sequencing decisions, as demonstrated through rapid phenotypic profiling of SARS-CoV-2 viruses. This method significantly increased the

efficiency of even sampling of reads in highly diverged samples such as supernatant and cellular RNAs in SARS-CoV-2 culture.

While WarpDemuX represents a notable advancement in nanopore direct RNA sequencing technology, it is essential to evaluate its practical applicability in the context of rapidly evolving sequencing methodologies. As with any innovative technique, certain challenges must be addressed to facilitate widespread adoption and ensure long-term relevance. The following points highlight areas where additional development or clarification could enhance the method's utility and accessibility for researchers.

Major points

1. As the authors already mentioned in the discussion, Oxford Nanopore Technologies replaced SQK-RNA002 with SQK-RNA004 in January 2024. SQK-RNA002 has been unavailable commercially for over six months now. To use WarpDemuX with the current kit, users must synthesize numerous long DNA-RNA hybrid reference molecules for training, then train and validate the model themselves. This requirement likely limits adoption to only a few researchers worldwide. To ensure the manuscript's relevance and impact, providing an updated SVM-DTWD kernel is crucial. Without this update, the paper's value to the field may be limited to historical documentation.

2. Figure 2c–d: While WarpDemuX significantly outperforms DeePlexiCon, its accuracy may still be insufficient for general gene expression analysis. In DNA sequencing, barcode accuracy routinely exceeds ~99.9%. The authors should discuss potential methods to improve performance, such as lengthening the barcode or implementing a more sensitive classifier algorithm.

3. Figure 2c–d: The authors should evaluate and describe the uneven misclassification of barcodes. Barcodes with similar signal patterns may have a higher chance of cross-contamination. Unlike classification based on basecalls, the cross-contamination tendency in signal-based classification can be complex and unpredictable for average users. This uneven cross-contamination may lead to artifactual conclusions in differential gene expression analysis. Figures SN.2.2 and SN.2.4 clearly show the uneven distance between barcodes. The authors should evaluate how this problem may affect downstream analysis, suggest methods to mitigate this issue, and ideally, propose a method to achieve comparable cross-contamination rates between any two barcodes at a single WDX confidence threshold.

4. Line 192: The benefits of adaptive sampling for direct RNA sequencing may be limited. Unlike genomic DNA fragments, eukaryotic mRNA molecules generally finish translocation within ~10 seconds even with their full lengths. The authors should thoroughly assess whether adaptive sampling effectively increases the number of reads from low-abundance samples. They should address the noticeable losses in channels undergoing adaptive sampling, as shown in Figure SN.4.3, which warrants further clarification in the text. Additionally, they should compare the effectiveness of adapter balancing in increasing read counts from low-abundance samples, rather than focusing solely on differences in relative compositions between barcodes.

5. Supplementary material page 47 (only in bioRxiv version): Regarding the reduction of chunk size, this smaller chunk sizes may lead to signal normalization issues, potentially affecting segmentation and pore state classification. ONT's signal acquisition software uses local statistics to detect stalls or blockages, so short chunk sizes may introduce more problems than benefits. The authors should conduct experiments to carefully evaluate any differences in overall quality, length distribution, and output when this modified configuration is introduced.

Minor Points

1. Figure SN.4.2: The terms "read_balancing" and "base_balancing" are used in the figure without explanation. The authors should provide clear definitions or descriptions of these terms to ensure readers understand their significance in the context of the study.

2. Figure 4c–d: The current plots may overestimate the efficiency of barcode balancing for mRNA molecules. Since mRNAs are typically short, the estimation of latency for rejection should account for the total time from when the molecule enters the pore until its final rejection. The plots currently suggest that barcode balancing can reject unwanted reads within 100ms, which may be misleading. The authors should revise these figures to accurately represent the full process time for mRNA rejection.

3. Figure 1d–f: These panels are not referenced in the main text.

4. Figure 3: Some panels in Figure 3 appear to be misreferenced in the text, with the manuscript referring to non-existent panels.

(Remarks on code availability)

Reviewer #3

(Remarks to the Author)

I read with interest the study by van der Toorn and colleagues about "Demultiplexing and barcode-specific adaptive sampling for nanopore direct RNA sequencing". The authors present WarpDemuX, a novel machine learning algorithm, which offers an adapter-barcoding and demultiplexing approach for nanopore direct RNA sequencing (dRNA-seq), which operates directly on the raw signal and does not require basecalling. It significantly improves speed and accuracy through fast processing of raw data. Its utility is demonstrated through profiling of SARS-CoV-2 viruses, enabling identification of differences in transcript abundance and poly(A) tail lengths. Overall, WarpDemuX may represent a powerful and cost-effective multiplexing solution for nanopore dRNA-seq, whose further use has lagged behind due to high associated costs and inability to process multiple samples simultaneously. The authors also demonstrate real-time dRNA-seq demultiplexing and barcode-specific adaptive sampling, an application previously not available for dRNA-seq.

The manuscript is well presented. The figures and tables are well prepared. The methodological approaches are elegant (e.g. engineering dual RTA barcodes to provide a ground-truth labelling).

My main suggestions are as follows:

- A) Costs. The authors mention saving costs due to the possibility of multiplexing. Yet, dRNA-seq is known to produce fewer reads per flow cell as compared to normal genomic DNA sequencing with ONT, due to the lower translocation speed that needs to be used with the former. Thus, the question is whether the multiplexing would offer enough reads per sample for users to be cost efficient, e.g. for a shotgun metatranscriptomic analysis containing few target molecules where millions of reads would be needed (i.e. well beyond the few thousand reads produced in the study as a proof-of-concept in Table 1). The authors should provide more information on the cost calculation and expected reduction, especially in the context of the maximal data yield expected with the current technology on Flongle or normal MinION flow cells. A Flongle is cheap but cannot sequence more than a couple thousand reads. So it may be useful for a proof of concept, but not for production.
- B) Technological obsolescence. The authors conclude their abstract by stating "In summary, WarpDemuX is a broadly applicable". Due to the fast pace of technology development at ONT, the laboratory method for dRNA-seq and flow cell type (R9.4.1) are no more available to customers on the market. So the proposed wet laboratory method and data production cannot be reproduced experimentally. The previous supplies have been replaced by new kits and chemistry, which are currently in early access to customers, as indicated correctly by the authors in the Discussion line 263 onward. Thus, the work presented in this study is not sure to be applicable to the soon available dRNA-seq technology. If the current study is published, one would want to know if the method applies to the newly available technology and not to something from the past. If possible to have access to those new flow cells and chemistry, I would strongly suggest adding few experimental data to validate the method on the R10.4.1 chemistry too.
- C) Figures' resolution should be improved, as well as the font size which is often very small for easy reading on a printed version of the manuscript.
- D) The provided GitHub code should provide a minimal test example to demonstrate the expected input and output files, as most of the study deals with the software capabilities.

Alban Ramette

(Remarks on code availability)

I could install the software successfully, by following the instructions provided at:
<https://github.com/KleistLab/WarpDemuX>

While the following command worked fine (the menu could be printed to the console):
> warpdemux demux -h

The suggested tests on the aforementioned GitHub page, namely:

```
>python -m warpdemux.live_balancing.dummy
```

or

```
>python -m warpdemux.live_balancing.dummy --config_file warpdemux/live_balancing/test/config_only_adapter_count.toml
```

produced the following error: "ModuleNotFoundError: No module named 'grpc'".

Beside this Module error, further software testing would not be possible because no test data or study data has been made available.

Reviewer #4

(Remarks to the Author)

Dr. van der Toorn and colleague have provided for the field of direct RNA sequencing an accurate, cost effective, and reasonably simple to use sample multiplexing methodology. The advantage of SVM-DTWD classification for capturing barcode fingerprints was evident in the obtaining barcode signature regardless of the dwell-time variation. Moreover, the speed of capture was impressive. The identification of 12 optimized barcodes, and data provided, highlights the expanded benefits of the new barcodes, which are advancements for the field. Nice demonstration of adaptive sampling for dRNA-seq using poly(A) tails with the new methodology which is problematic with DPC.

The advantage of WarpDemuX is that it is ready to go for ONT SQK-RNA002, which is, unfortunately (for the authors), the discontinued and older ONT sequencing chemistry. The authors note that the changes in the sequencing chemistry will require retraining of the SVM-DTWD kernel for the new sequencing chemistry used in SQK-RNA004, the current ONT

standard. However, it is not clear that retraining of the SVM-DTWD kernel for the new ONT sequencing chemistry is going to be straightforward for the average user. Moreover, current WarpDemu barcodes, a major move forward in the field, may not work with the new ONT sRNA-specific pore display necessitating evaluation of current barcodes and possibly the development of new barcodes.

Unfortunately, the authors were caught in an ONT sequencing chemistry upgrade that puts into question the general adaptation of WarpDemuX and barcodes for the dRNA-seq field. These concerns need to be addressed by the authors.

(Remarks on code availability)
i did not load / run the code.

Reviewer #5

(Remarks to the Author)

(Remarks on code availability)

The code provides sufficient instructions for ad-hoc demultiplexing (after sequencing), and for live barcode balancing. The instructions and containers provided seem reasonable and well annotated for typical Nanopore users. However, training models provided are for RNA002 chemistries which have been deprecated and are not available anymore. Authors need to provide updated models OR provide a detailed README (along with associated tools and code) for users to train their own models using RNA004 chemistries which are currently available at Nanopore.

Version 1:

Reviewer comments:

Reviewer #1

(Remarks to the Author)

Thanks to the authors for responding to my comments about warpDemuX.

As I indicated in my initial review, I am broadly in favor of the approach. However, as I have continued to review it I am struggling to clearly and simply identify the basic fundamentals in what the authors are saying.

I suggested the authors performed a barnyard experiment to determine the accuracy of barcode classification. This was not simply to determine if there is cross talk from sequences alone, but is to ensure the broader general accuracy of the method. If one is barcoding - and further more making decisions that influence sequencing behavior - then precise confirmation of the accuracy of barcoding and demultiplexing is essential. The RTA barcode shuffling approach may confirm this, but I struggled to find this clearly presented in the manuscript. I cannot see how the authors can make claims about the truth state of the SARS CoV-2 data they have presented. The exclusion of unclassified reads is an unusual rationale. These reads must have barcodes and so the lack of classification is a consequence of the method, not the input.

Figure 4f is puzzling. What is the control group here? Is it the whole of the no WDX portion of the flowcell? Or is it the no balancing component? To properly evaluate the comparison these details need to be recorded. The results are not shown for the read count or the base count. I presume the comparison must be against the no balancing section. Please clarify.

The authors have modified the text about the issues around base calling and signal processing, but this text is still misleading.

The authors state that GPU is computationally expensive. An option that doesn't require GPU simply doesn't use GPU. If using Nanopore sequencing one needs to have GPU to enable base calling. Similarly the argument about the number of reads that can be processed per second is misleading. WarpDemuX can process a significant number of reads - 50,000 per minute. But, a MinION running at full speed on the assumption of 0.1 seconds of data per read can generate at max 5,120 reads. Thus the 10,000 reads/minute of SeqTagger is still more than sufficient. For a PromethION device, this isn't the case. But it is unclear if a promethION device can supply signal in chunks of 0.1 seconds. This also raises a false equivalence with the comparisons made between basecalling and signal analysis. In this work the authors have reduced the break_read_seconds parameter to 0.1 seconds. Prior work changed this to 0.4 seconds for base calling. The standard MinKNOW setting is 1 second. The revised text the authors present is incorrect.

(Line 199-207) "Although basecalling approaches effectively determine nucleotide sequences, they impose significant computational overhead and require the accumulation of a sizable amount of raw signal data before analysis can commence. This inherent latency introduces delays in adaptive sampling decision-making and allows unwanted molecules to traverse further through the pore before rejection. This extended pore occupancy negatively impacts pore lifetime and sequencing yield, reducing the effectiveness of the approach (Kovaka et al.). Signal-based approaches circumvent these limitations and can enable more rapid molecule rejection, preserving pore integrity and resulting in better overall enrichment

and/or yield of low abundance samples. Yet, this barcode balancing approach remains unexplored in the context of dRNA-seq.”

Basecalling works on as little as 40 bases of DNA - which equates to 0.1 seconds. To make the statements above that the authors claim they would need to test base calling performance at 0.1 second chunks. The inherent latency is not derived from basecalling - it is derived from the underlying chunk size switch. The authors quote Kovaka et al on impacts on pore lifetime, but this is again linked to the change in unblock time and nothing else. The signal-based approach accelerations that are reported here are equally achievable with basecalling approaches. After all, the basecalling method is itself a signal based approach. The authors do not demonstrate any of the claims they make about pore integrity, better overall enrichment or yield of low abundance samples. I do not disagree with the authors point that processing less data faster will result in better performance. I do disagree that this is an inherent property of the method presented here.

I would strongly suggest the authors consider reducing the complexity of the supplementary material and increasing the transparency of reporting. Figure SN.5.3 shows the percentage of reads rather than absolute numbers- absolute numbers would be a useful comparison.

(Remarks on code availability)

Reviewer #2

(Remarks to the Author)

I have carefully reviewed the revised manuscript and responses to the reviewers' comments. I am pleased to see how thoroughly and diligently the authors have addressed all the feedback.

The incorporation of RNA004 kit support significantly enhances the manuscript's relevance for current sequencing technologies. The authors' enhanced analysis of potential biases, expanded discussions, and additional validation data have substantially improved the scientific rigor of the work. The revised figures and supplementary materials provide excellent clarity to the work.

After thorough consideration, I believe the manuscript in its current form is suitable for publication.

(Remarks on code availability)

Reviewer #3

(Remarks to the Author)

I am pleased by the authors' modifications and changes to their manuscript.

(Remarks on code availability)

I could install the code and the test data have been provided this time.

The GitHub page is also well prepared and enables a new user to get familiar with the software.

Reviewer #4

(Remarks to the Author)

In the revised manuscript, the Authors now provide evaluation of WarpDemuX and barcodes using the current generation nanosequencing chemistry and have added a test example to the GitHub repository to demonstrate the expected input and output files. Figures have been improved. I am satisfied with the Authors' additions to the revised manuscript. The methodology could be useful for the field in the current form.

(Remarks on code availability)

Version 2:

Reviewer comments:

Reviewer #1

(Remarks to the Author)

I thank the authors taking the time to address my comments.

Comment 1.

The authors have explained the shuffling clearly enough for me to follow. The text changes are also clear in this regard.

Comment 2.

I agree with the authors that not all signals initially classified by MinKNOW represent genuine RNA molecules. I also agree with the issues of the difficulties of classification when you have a highly imbalanced data set. Of course setting a high threshold is important.

My question here is why are the unclassified reads not explored. For example, of the 363,283 reads that pass a Q score filter, 16592 reads do not have barcodes. Is that because they are completely missing - i.e they have no barcode - or is the barcode there, but not detectable by warpDemux? The reads that are subsequently filtered out (109,739 reads) all have barcodes but - as pointed out by the authors - have a lower classification confidence. The authors state (lines 163-166) that the correlation between barcodes and EGFP sequences is effectively perfect in the reads that pass classification. Does this correlation remain in the 109,000 reads that can be assigned a barcode but which - quite rightly - may be excluded from the subsequent analysis for other reasons.

Line 160 is missing a specific reference to a figure.

Comment 3

Thanks to the authors for providing extra detail. The new figure presented in supplementary figure 5.3 is helpful, but is not referenced in the main text. I presume that by "passing" read counts and "passing" read bases the authors are referring to the data that passes the thresholds already discussed. The real question is if the method results in more absolute data on target. The relative proportions shown in SN.5.3 a and b show that the data are normalized, but do not show the absolute enrichment. This is shown in SN 5.4. The figure I was hoping to see was a panel similar to SN.5.4 but with the total bases sequenced (absolute number of bases barcode - not relative numbers) as well as a count of the total number of reads and bases unclassified in each balancing strategy. This would unambiguously demonstrate the enrichment and removes any ambiguity from the data.

Comment 4 (a).

The authors fail to mention that the recommended GPU compute - the RTX4090 - is sufficient for both real time base calling and implementation of adaptive sampling with respect to DNA. Given the higher sequencing speed of DNA sequencing to RNA it is ergo simpler to run the analysis on a GPU. It is also not correct to assume that CPU resource is unlimited in these scenarios. It is very easy to overwhelm the CPU requirements whilst running live code alongside minKNOW. The only reference I can find to compute resource is an Intel i7-1165G which is the minimum requirement for running a minION. This is good performance, but is this the set up the authors used for their experiment?

I completely agree with the authors that efficient and effective code is vital. I disagree that there is evidence to say that this is a practical limitation.

Comment 4 (b)

The post-run scenario the authors describe is not really likely. I come back to my point that GPU base calling is - really - an essential pre-requisite to any laboratory running nanopore sequencing at scale. A laboratory using multiple promethION flowcells is likely to be able to implement GPU calling. Nonetheless I do take the point that post run demultiplexing rapidly is good.

Comment 5

Re-optimising the sequencing toml files and configuring these for short chunks is a lot of work. I don't understand the comment at line 1100 in the supplementary material - "Unfortunately, these necessary changes resulted in performance highly similar to the default protocol" - why is this unfortunate? In figures sn 7.2 and 7.3 are all of these data tuned on simulated runs (playback) or did the authors try switching between protocols on a real sequencing run to determine no negative effects. The requirement to run a modified sequencing script is something that should be made very clear in the main text? I cannot find this anywhere?

My comment about the text being misleading is that - for adaptive sampling - the fact that the break reads parameter does not have to be 1 has been previously reported. If the authors have resolved the toml files for this then have they tested the performance of inbuilt adaptive sampling using base calling against their method? I do realise that this is almost certainly impossible in the context of this specific experiment as the integration of the barcode into the adapter precludes this. But it would be of interest to know if the performance improvements translate. In addition, is the base calling of the data impacted in anyway by the change in sequencing parameters - one presumes not?

Comment 6

The authors have made a number of assumptions about how the base calling process is configured and how "batching" occurs. The models used do not require batching - reads can be sent as singletons or as multiples. The authors have no doubt noticed that the reads they receive via the read until API are themselves in batches. My point was that the authors could test performance of base calling at 0.1 seconds and determine if the improvements come from their optimized algorithms or simply theft that data are now available in 0.1 second bursts. In the authors comparison of the approaches they state the latency is low since each read is immediately processed. The same can be achieved with base calling. So - the

claim the authors make with respect to latency and the requirement for waiting for enough reads to batch is incorrect. A single read can be sent (and is sent further down).

The authors have addressed my concerns with respect to running some simulations, but they do so within the framework of readfish which, as I understand it, inherently batches reads rather than sending them asynchronously. The authors state their own model calls one read every 43 Ms. I couldn't immediately find the data for that but I do not disbelieve it. It would be worthwhile reporting the mean time as well as the median.

The authors test more complex models (hac etc) and I agree that the performance drops here. The authors state they test HAC as this gives equivalent performance to WarpDemuX - how is this shown?

The authors quote ONT's notes that say that necessary compute is required for adaptive sampling to run successfully. I agree with this. I also agree with the shown figures that ONT's N50 at rejection is around 680 bases. And I agree with the authors calculation of the approximate latency in the system (0.7 seconds) after data collection.

What are the equivalent values for the code in this work running on an entire flowcell (not a small section thereof). What are the values when running a similar base calling experiment using the 0.1 second unblock time?

I note that the authors state "While we acknowledge that further accelerations could theoretically be achieved by optimized basecalling approaches, we trust the reviewer can appreciate that in this work we did not aim to make direct comparisons to such theoretical alternatives.". Indeed - I agree with this. But it is the authors who are not acknowledging this in the manuscript text.

I really think the approach developed by the authors is innovative and of interest. I note the comment on line 305 of the manuscript that the approach is independent of base calling. This is a really neat trick, but as the authors acknowledge requires new SVM-DTWD kernels if the chemistry changes. If one had an RNA barcode that could be base called, how would this compare?

Comment 7.

The authors point that they aim to unblock reads whilst in the poly(A) tail reduces pore blocking is well shown. They also indicate their method has no negative impact on pore life span. These data are excellent. The issue with the base calling approach is more nuanced. If the authors wish to test this they should unblock every read on a flow cell (using base calling) and look at the loss of pores compared to a control region. The difference is imperceptible and only really emerges 24 hours into a run. I do not know how long it takes on RNA. I would imagine that RNA might be more sensitive to blocking - but the authors do not show this.

The authors have modified their statement, but really the point here is that they do not show the impact of base calling adaptive sampling on RNA sequencing performance and so they cannot make a comparison.

Comment 8,

I am sorry it has taken a long time to respond to the authors.

This is - in part - because the supplementary notes, explanations and figures are now at 65 pages. Much of this is to support the authors relative comparison claims.

As I understand it, the approach that the authors present cannot use base calling. So it isn't an option. The solution is elegant and works well - providing a good and efficient RNA barcoding approach. This is what the manuscript should be emphasizing (in my view). The adaptive sampling data shown in figure sn 5.4 show a less than 2 fold improvement in the number of reads from the least abundant barcode from an awful lot of work. In my view this is the least exciting aspect of the paper - far more exciting is the ability to efficiently barcode RNA samples.

(Remarks on code availability)

I have inspected the code but I have not been able to run it as it requires generation of a sequencing library to test properly.

Version 3:

Reviewer comments:

Reviewer #1

(Remarks to the Author)

The authors have largely addressed many of my comments.

The authors incorrectly state that base calling approaches to RNA adaptive sampling are not available - for example see Wang, J., Yang, L., Cheng, A. et al. Direct RNA sequencing coupled with adaptive sampling enriches RNAs of interest in the transcriptome. Nat Commun 15, 481 (2024). <https://doi.org/10.1038/s41467-023-44656-3>

In my view the only way to really test adaptive sampling experiments is by running a real experiment on a sequencing flow cell. Whilst one can simulate with playback and other simulations it is not the same as running an actual experiment. Similarly, it is not correct to assume that performance will scale linearly (presumably inversely) with pore count - as minKNOW itself is adding complexity to the process.

I remain of the view that the true innovation here is the excellent barcoding approach and that the benefits of adaptive sampling on RNA are minimal.

(Remarks on code availability)

Dear Reviewers,

We sincerely thank you for taking the time to review our manuscript and for providing valuable feedback. Your comments have helped us improve the quality and clarity of our work significantly.

In particular, we would like to highlight that we have conducted additional RNA sequencing experiments (RNA004 dataset) in response to the reviewers' comments. The experiments allowed us to adapt and further improve the previous computational workflow to the latest dRNA-seq chemistry SQK-RNA004, with novel results reported in the new section "*Adaptation to RNA004 chemistry maintains high performance*" and Fig. 5.

We hope that our revisions have adequately addressed all the concerns raised. Below, we address each comment point by point.

Reviewer #1

Experiment Suggestion

Comment: The only experiment I'd like to see in an ideal world is a "barnyard" type experiment where barcodes were used on RNA derived from different species. This would allow the species to act as the control barcode and thus the possibility of specific sequences interfering with the barcode detection could be tested. However - this is not an absolute requirement - but would be nice to see!

Response: We appreciate the reviewer's emphasis on addressing sequence-specific biases in direct RNA sequencing.

Our barcode detection system is inherently robust against sequence-specific interference because it relies on identifying the adapter/poly(A) boundary prior to barcode classification, with the poly(A) tail serving as a signal buffer between the adapter and any RNA sequence. Nevertheless, to systematically control for potential sequence biases, we shuffled RTA barcode assignments across different combinations of inline RNA barcodes and RNA species between replicates. Additionally, we have demonstrated our method's effectiveness on SARS-CoV-2 transcripts, a completely different RNA species, with strong performance - further validating that sequence-specific biases do not impact our results. While a multi-species "barnyard" experiment could provide additional validation, our current experimental design and cross-species validation effectively address these concerns.

Speed Gain Claim

Comment: The authors cite Kovaka et al as evidence that signal based approaches are faster for read classification than base calling approaches. This claim is not supported by the work

of Kovaka et al when considered in context. The paper from Payne et al that was published back-to-back with Kovaka et al demonstrates that base calling outcompetes signal based approaches in these contexts. I was going to ask why the authors had not compared their methods with direct basecalling approaches but then realized the elegance of the approach of using DNA barcodes in an RNA sequencing context. Therefore I suggest that the authors simply de-emphasize these comments.

Response: Thank you for your comment. We have revised our discussion of basecalling- v.s. signal-based decision-making in adaptive sampling. We now focus on the inherent characteristics of basecalling-based methods that were discussed by Kovaka et al. Specifically, we highlight the computational overhead and the requirement for accumulating sufficient raw signal data before analysis can begin. We have reframed the citation of Kovaka et al. to focus on these general challenges of basecalling-based approaches. In addition, we have added an extra analysis addressing the latency and speed of WarpDemuX in the context of adaptive sampling efficiency, showing that WarpDemuX approaches the theoretical maximum efficiency (zero latency and instantaneous classification) for transcripts of 300 nucleotides and longer (Supplementary Notes SN5.3 section “*Efficiency of WarpDemuX adaptive-sampling*” and Supplementary Figure SN.5.2). We would also like to note that a basecalling based demultiplexing approach was presented last week, in which the authors developed a custom RTA basecaller (“SeqTagger”) that reads out the specific nucleotide sequences of DNA barcodes placed in the RTA adapter (<https://doi.org/10.1101/2024.10.29.620808>). For RNA004, the authors present performance metrics on a 4-barcode model, which are comparable with WDX4 performance, yet require significant computational overhead through GPU acceleration while being five times slower, processing approximately 10,000 reads/minute compared to WDX4’s 50,000 reads/minute (SeqTagger Figure 2 vs WarpDemuX Figure 4g).

Changes:

PREVIOUS “However, basecalling is slow, allowing unwanted molecules to traverse a substantial distance through the pore before rejection, which negatively impacts pore lifetime and sequencing yield. Making adaptive sampling decisions on raw signal can be significantly faster (Kovaka et al.), preserving pore integrity and resulting in better overall enrichment and/or yield of low abundance samples. Yet, this barcode balancing approach has not yet been implemented for dRNA-seq.”

REVISED (Line 199-207) “*Although basecalling approaches effectively determine nucleotide sequences, they impose significant computational overhead and require the accumulation of a sizable amount of raw signal data before analysis can commence. This inherent latency introduces delays in adaptive sampling decision-making and allows unwanted molecules to traverse further through the pore before rejection. This extended pore occupancy negatively*

impacts pore lifetime and sequencing yield, reducing the effectiveness of the approach (Kovaka et al.). Signal-based approaches circumvent these limitations and can enable more rapid molecule rejection, preserving pore integrity and resulting in better overall enrichment and/or yield of low abundance samples. Yet, this barcode balancing approach remains unexplored in the context of dRNA-seq."

Benchmarking Against Basecalling

Comment: The authors should clarify that their method cannot be benchmarked against base calling due to the mixed model of DNA and RNA. It should also be clarified that if RNA barcodes were made available by ONT then this experiment would be possible.

Response: We would like to clarify that the fact that basecalling of the RTA is not possible with the available RNA basecaller is addressed in the introduction. We have included the suggested comment on the technical infeasibility of benchmarking basecalling- and signal-based demultiplexing approaches in the discussion. We would also like to note that a basecalling based demultiplexing approach was presented last week, in which the authors developed a custom RTA basecaller ("SeqTagger") that reads out the specific nucleotide sequences of DNA barcodes placed in the RTA adapter (<https://doi.org/10.1101/2024.10.29.620808>). For RNA004, the authors present performance metrics on a 4-barcode model, which are comparable with WDX4 performance, yet require significant computational overhead through GPU acceleration while being five times slower, processing approximately 10,000 reads/minute compared to WDX4's 50,000 reads/minute (SeqTagger Figure 2 vs WarpDemuX Figure 4g).

Changes:

REVISED (Line 271-276) *"In all these approaches, DNA barcodes embedded in the standard (DNA-based) sequencing adapter are used to demultiplex RNA sequences. The RNA basecaller's inability to process DNA barcodes limits direct comparisons between signal-level analysis and basecalling-based methods.. Should ONT introduce RNA-based barcodes in their sequencing protocols, for example through the use of RNA-based sequencing adapters, it would enable comprehensive benchmarking studies comparing signal-based and basecalling approaches, while also allowing investigation of potential library preparation biases between the use of DNA and RNA barcodes. Additionally, such studies could investigate the risk of overfitting in basecalling models, given the relatively small sequence space of barcodes compared to full-length RNA sequences. Whilst this manuscript was in review, Prysycz et al., developed a custom RTA basecaller ("SeqTagger") that reads out the specific nucleotide sequences of DNA barcodes placed in the RTA adapter (Prysycz et al, 2024 BioRxiv). For RNA004, the authors present performance metrics on a 4-barcode model, which are comparable with WDX4 performance, yet require significant computational*

overhead through GPU acceleration while being five times slower, processing approximately 10,000 reads/minute compared to WDX4's 50,000 reads/minute (Fig.5g). “

WarpDemuX Classification

Comment: If I have followed the paper correctly, the warp demux approach only allows classification based upon barcode and not based upon a specific molecule within the sample. If this is not the case, it would be nice to clearly see that shown in the manuscript.

Response: Indeed, the WarpDemuX approach was developed for the classification of (RTA) barcodes attached to RNA molecules (one barcode per sample / condition), not the classification of specific molecules or types of RNA within the sample.

Reviewer #2

Relevance and Impact

Comment: As the authors already mentioned in the discussion, Oxford Nanopore Technologies replaced SQK-RNA002 with SQK-RNA004 in January 2024. SQK-RNA002 has been unavailable commercially for over six months now. To use WarpDemuX with the current kit, users must synthesize numerous long DNA-RNA hybrid reference molecules for training, then train and validate the model themselves. This requirement likely limits adoption to only a few researchers worldwide. To ensure the manuscript's relevance and impact, providing an updated SVM-DTWD kernel is crucial. Without this update, the paper's value to the field may be limited to historical documentation.

Response: We appreciate the point raised and have addressed it by developing the model for the new chemistry. We provide the results of an updated SVM-DTWD kernel to ensure the manuscript's relevance and impact. This is documented in the new section “*Adaptation to RNA004 chemistry maintains high performance*”, new Fig. 5 and corresponding updates to the code repo.

Accuracy Improvement

Comment: Figure 2c–d: While WarpDemuX significantly outperforms DeePlexiCon, its accuracy may still be insufficient for general gene expression analysis. In DNA sequencing, barcode accuracy routinely exceeds ~99.9%. The authors should discuss potential methods to improve performance, such as lengthening the barcode or implementing a more sensitive classifier algorithm.

Response: The reviewer raises an important point about accuracy requirements. We have further improved the performance of WarpDemuX while adapting the model to the new RNA004 chemistry. We believe that the WarpDemuX >99% demultiplexing performances

provide a sufficient solution for most use cases. Additionally, we further improved the way in which the user can balance the trade-off between accuracy and yield, providing a solution for use cases in which ~99.9% demultiplexing accuracy is required (new Methods section “RNA004 Confidence Score Calibration”, Table 2, and the new Supplementary Notes SN.3 “WarpDemuX performance RNA004”), i.e. sensitivity can be tuned now.

Note that currently the length of the standard RTA barcode is 18nt, which poses a theoretical limit to the search space of potential barcode sequences with distinct signal patterns. However, in principle it is possible to create adapters with longer RTA barcodes. We have added a sentence to the discussion to address the benefits of lengthening the barcode.

Changes:

REVISED: (Line 292-294) “Lengthening the barcodes could effectively push back this saturation point, allowing for the implementation of substantially larger barcode sets before reaching capacity limitations while enhancing barcode classification confidence for smaller sets.”

Uneven Misclassification

Comment: Figure 2c–d: The authors should evaluate and describe the uneven misclassification of barcodes. Barcodes with similar signal patterns may have a higher chance of cross-contamination. Unlike classification based on basecalls, the cross-contamination tendency in signal-based classification can be complex and unpredictable for average users. This uneven cross-contamination may lead to artificial conclusions in differential gene expression analysis. Figures SN.2.2 and SN.2.4 clearly show the uneven distance between barcodes. The authors should evaluate how this problem may affect downstream analysis, suggest methods to mitigate this issue, and ideally, propose a method to achieve comparable cross-contamination rates between any two barcodes at a single WDX confidence threshold.

Response: This is a very good point, thank you for raising it. To address this, we implemented class-specific confidence filtering for predefined target accuracies for our RNA004 models. From the newly sequenced RNA004 data, we kept 2000 reads per barcode aside for calibration. We enhanced our barcode classifier models with an additional post-training calibration step to achieve comparable cross-contamination rates between any two barcodes given a predefined target accuracy (95% to 99.9%). The details of the class-specific confidence scores are discussed in the Methods section “RNA004 Confidence Score Calibration” and the results are shown in Figure 5 and Supplementary Figure SN3.1-3.4, and Supplementary Table SN3.1 and Supplementary Table SN3.2.

Adaptive Sampling

Comment: Line 192: The benefits of adaptive sampling for direct RNA sequencing may be limited. Unlike genomic DNA fragments, eukaryotic mRNA molecules generally finish

translocation within ~10 seconds even with their full lengths. The authors should thoroughly assess whether adaptive sampling effectively increases the number of reads from low-abundance samples. They should address the noticeable losses in channels undergoing adaptive sampling, as shown in Figure SN.4.3, which warrants further clarification in the text. Additionally, they should compare the effectiveness of adapter balancing in increasing read counts from low-abundance samples, rather than focusing solely on differences in relative compositions between barcodes.

Response: We wholeheartedly agree with the reviewer that use of adaptive sampling needs to be considered carefully, especially with shorter input molecules such as RNA. Considering this shorter read length we performed in depth optimization to reduce latency of rejection, resulting in an average time from molecule start to rejection of only 0.3 seconds. This resulted not only in more balanced read abundance (Supplementary Figure SN5.3), but indeed increased the absolute number of reads for low abundance barcodes (Fig 4f). We would like to point out that Supplementary Figure SN5.4 (previously, SN4.3) shows an analysis of pore lifetime (green area) for channels with different adaptive sampling strategies compared to channels without adaptive sampling (left-most panel; strategy 'none'). The observable loss in sequencing channels over time (decrease of the green area) that the reviewer correctly points out is an inherent characteristic of nanopore sequencing, and occurs independent of adaptive sampling. We found no substantial differences in lifetime dynamics between pores with and without adaptive sampling, regardless of the balancing strategy used. The reason why we do not observe this effect may be attributed to our rejections occurring within the poly A tail, which means there is no RNA on the trans side of the pore potentially forming stable secondary structures. We have improved the figure caption to clarify the analysis.

Chunk Size Reduction

Comment: Supplementary material page 47 (only in bioRxiv version): Regarding the reduction of chunk size, this smaller chunk sizes may lead to signal normalization issues, potentially affecting segmentation and pore state classification. ONT's signal acquisition software uses local statistics to detect stalls or blockages, so short chunk sizes may introduce more problems than benefits. The authors should conduct experiments to carefully evaluate any differences in overall quality, length distribution, and output when this modified configuration is introduced.

Response: This is indeed an important consideration. While altering the chunk size parameter we noticed the effects of reducing the chunk size on pore state classification, in particular in relation to truncated reads and detection of pore blockages. Based on several iterations, including simulated bulk file playback analyses and experimental runs, we developed our custom minKNOW protocol with adapted block detection/unblocking logic (included on the associated GitHub repo). We have evaluated this protocol to ensure it does not generate truncated reads, and is able to detect blocked pores to a similar extent as the default protocol.

We have added a paragraph and two supplementary figures, SN.7.2 and SN.7.3 containing a comparison of the default RNA002 protocol with our protocol on a simulated run.

In addition, the customized protocol was used to generate the data shown in Figure 3 and Figure SN.4.7, showing its ability to consistently sequence full length RNA molecules. We realized that this information was missing in the previous version of the document, and thank the reviewer again for raising awareness to this crucial point. We have updated the method section to include this detail.

We agree with the reviewer that altering the chunk size needs to be carefully considered. We have added a section highlighting the complexity involved in altering this parameter.

Figure SN4.2

Comment: The terms "read_balancing" and "base_balancing" are used in the figure without explanation. The authors should provide clear definitions or descriptions of these terms to ensure readers understand their significance in the context of the study.

Response: We have addressed this by explaining the terms in the figure caption (now Suppl. Fig. SN5.3).

Figure 4c–d

Comment: The current plots may overestimate the efficiency of barcode balancing for mRNA molecules. Since mRNAs are typically short, the estimation of latency for rejection should account for the total time from when the molecule enters the pore until its final rejection. The plots currently suggest that barcode balancing can reject unwanted reads within 100ms, which may be misleading. The authors should revise these figures to accurately represent the full process time for mRNA rejection.

Response: The total time until rejection in relation to the total duration of sequencing in the absence of rejection is indeed the crucial parameter in evaluating maximal enrichment efficiency. We agree with the reviewer that a plot showing the total time until rejection, including the translocation time of the adapter, will add to the interpretability of our adaptive barcode balancing strategy. We have included a detailed analysis of the total time from when the molecule enters the pore until its final rejection and efficiency of real-time adaptive sampling in the revised manuscript in Supplementary Notes, section SN.5 (*Efficiency of WarpDemuX adaptive-sampling*). By analyzing translocation speeds and the decision interval (time from pore entry to adaptive-sampling decision), we found that adapter regions translocate 1.3-fold slower than RNA segments (55.7 vs 70.8 nt/s). The adapter sequencing phase dominates the decision interval (84.7%), while WarpDemuX latency and processing adds minimal overhead (14.6% total). For templates exceeding 300 nucleotides, adaptive

sampling saves more than 50% of sequencing time in over 90% of rejected reads, with efficiency increasing for longer templates (Suppl. Fig. SN 5.2). This analysis furthermore showed that WarpDemuX performance closely approaches the theoretical maximum enrichment possible for RTA-based adaptive sampling, despite the real-world operational constraints. The metrics shown in Figure 4c and d (the duration of WarpDemuX processing, and the number of nucleotides sequenced between start of poly(A) and rejection request) are characteristics of our method, while the time required to sequence the adapter is inherent to direct RNA sequencing and independent of the method of adaptive sampling. We therefore have chosen not to update the metrics in the main figure.

Figure 1d–f

Comment: These panels are not referenced in the main text.

Response: Thanks for pointing this out, we have now referenced these panels in the text.

Figure 3

Comment: Some panels in Figure 3 appear to be miss referenced in the text, with the manuscript referring to non-existent panels.

Response: Thanks for pointing this out, we have fixed the references to the panels in the text.

Reviewer #3

Costs

Comment: The authors mention saving costs due to the possibility of multiplexing. Yet, dRNA-seq is known to produce fewer reads per flow cell as compared to normal genomic DNA sequencing with ONT, due to the lower translocation speed that needs to be used with the former. Thus, the question is whether the multiplexing would offer enough reads per sample for users to be cost efficient, e.g. for a shotgun metatranscriptomic analysis containing few target molecules where millions of reads would be needed (i.e. well beyond the few thousand reads produced in the study as a proof-of-concept in Table 1). The authors should provide more information on the cost calculation and expected reduction, especially in the context of the maximal data yield expected with the current technology on Flongle or normal MinION flow cells. A Flongle is cheap but cannot sequence more than a couple thousand reads. So it may be useful for a proof of concept, but not for production.

Response: We have sequenced our RNA004 dataset using a PromethION flow cell, showing WDX is flow cell-independent and can also be applied in production settings. We agree that multiplex sequencing may not be viable for use cases where exceptionally deep coverage is required. For all other use cases however, the benefits of pooled sequencing of barcoded

samples has been well-established and can reduce costs by a magnitude. Important to note is that ONT is not providing a Flongle flow cell for the new RNA004 chemistry, as such every single sample needs to be sequenced on a Minion or Promethion flow cell, with multiple rounds of washing, substantially increasing hands-on-time and cost per sample. Our multiplexing strategy reduces the cost per sample below single Flongle flow cells per sample, while increasing the output per sample, as Promethion flow cells have approximately 50x more yield (practical yield for RNA: Flongle 1 Gb, PromethION 50 Gb) .

Below (Table 1-4), we've included more detailed calculations on costs and expected reduction, also in the context of the maximal data yield expected. We have now also added these calculations to the manuscript as an Appendix (Supplementary section SN.8). Values were calculated using the following estimates:

- Practical yield of MinION flowcell, RNA: 10 Gb
- Practical yield of PromethION flowcell, RNA: 50 Gb
- Number of times a flowcell can be washed: 5 times
- Maximal number of samples to be multiplexed together on a single flowcell: 24 samples
- Yield of demultiplexing workflow: 85%
- Costs of Library Preparation Kit: €600 (6 runs per kit)
- Costs of flowcell Wash Kit: €100 (6 runs per kit)
- Costs of flowcell: €800
- Cost of Labor: €60,00/hour
- Handson-time required: 30 minutes for washing flow cell, 1 hour for preparing library and 15 minutes for flowcell loading, per flowcell

Number of samples	Number of flowcells	Number of washes	Expected yield per sample (Gb)	
			MinION (10Gb)	PromethION (50Gb)
2	1	1	5,0	25,0
4	1	3	2,5	12,5
6	2	4	3,3	16,7
8	2	6	2,5	12,5
10	2	8	2,0	10,0
12	3	9	2,5	12,5

Table 1. Expected maximal yield per sample, without sample multiplexing, assuming flowcells can be washed (reused) 5 times.

Number of samples	Number of flowcells	Number of washes	Expected yield per sample (Gb)	
			MinION (10Gb)	PromethION (50Gb)
2	1	0	4,3	21,3

4	1	0	2,1	10,6
6	1	0	1,4	7,1
8	1	0	1,1	5,3
10	1	0	0,9	4,3
12	1	0	0,7	3,5

Table 2. Expected maximal yield per sample, with sample multiplexing, assuming yield of demultiplexing to be 85%.

Number of multiplexed samples	Absolute savings compared to non-multiplexing (flow cell washing)			
	Cost per Sample	Cost per Experiment	Cost per Gb	Time (h:mm)
2	79 €	159 €	-1,64 €	2:15
4	112 €	450 €	4,17 €	5:15
6	252 €	1.510 €	0,59 €	4:00
8	225 €	1.801 €	3,47 €	6:00
10	209 €	2.092 €	6,36 €	8:00
12	263 €	3.152 €	3,17 €	7:25

Table 3. Absolute savings of sample multiplexing compared to sequencing without multiplexing.

Number of multiplexed samples	Relative savings compared to non-multiplexing (flow cell washing)			
	Cost per Sample	Cost per Experiment	Cost per Gb	Time
2	14%	14%	-7%	43%
4	32%	32%	15%	64%
6	61%	61%	2%	57%
8	65%	65%	13%	67%
10	68%	68%	21%	73%
12	76%	76%	12%	71%

Table 4. Relative savings of sample multiplexing compared to sequencing without multiplexing.

Technological Obsolescence

Comment: The authors conclude their abstract by stating "In summary, WarpDemuX is broadly applicable". Due to the fast pace of technology development at ONT, the laboratory method for dRNA-seq and flow cell type (R9.4.1) are no longer available to customers on the market. So the proposed wet laboratory method and data production cannot be reproduced experimentally. The previous supplies have been replaced by new kits and chemistry, which are currently in early access to customers, as indicated correctly by the authors in the Discussion line 263 onward. Thus, the work presented in this study is not sure to be applicable

to the soon available dRNA-seq technology. If the current study is published, one would want to know if the method applies to the newly available technology and not to something from the past. If possible to have access to those new flow cells and chemistry, I would strongly suggest adding a little experimental data to validate the method on the R10.4.1 chemistry too.

Response: We appreciate the point. To stay true to our statement and provide a useful tool for future research, we have developed and validated models for the new RNA004 chemistry, as elaborated in the new section "*Adaptation to RNA004 chemistry maintains high performance*" and showcased in the new Fig. 5.

Figures' Resolution

Comment: Figures' resolution should be improved, as well as the font size which is often very small for easy reading on a printed version of the manuscript.

Response: Thank you for the suggestion, we have made adjustments to the figure fonts to improve readability.

GitHub Code

Comment: The provided GitHub code should provide a minimal test example to demonstrate the expected input and output files, as most of the study deals with the software capabilities.

Response: We have added a test example to the GitHub repo to demonstrate the expected input and output files. Thank you for pointing this out.

Reviewer #3 (Remarks on code availability)

Module Error

Comment: The suggested tests on the aforementioned GitHub page produced the following error: "ModuleNotFoundError: No module named 'grpc'". Beside this Module error, further software testing would not be possible because no test data or study data has been made available.

Response: Thank you for pointing this out, we have fixed the live_balancing test, updated the environment, and added data to the GitHub repo to resolve the module error and facilitate further software testing.

Reviewer #4

Evaluation of Current Barcodes

Comment: ... current WarpDemu barcodes, a major move forward in the field, may not work with the new ONT sRNA-specific pore display necessitating evaluation of current barcodes and possibly the development of new barcodes. Unfortunately, the authors were caught in an ONT sequencing chemistry upgrade that puts into question the general adaptation of WarpDemuX and barcodes for the dRNA-seq field. These concerns need to be addressed by the authors.

Response: We appreciate the point and developed signal processing parameters and models that work with the new sequencing chemistry. We have added experimental data to support our findings in the revised manuscript, in particular the new section "*Adaptation to RNA004 chemistry maintains high performance*" and Fig. 5. Indeed, with the changes to the signal properties, we have found that some of the selected WarpDemuX barcodes are no longer optimal and the further development, optimization and expansion of RNA004 barcode sets will be the subject of future work. However, using the current setup, we can already demonstrate convincing performance on WDX4 and WDX10 in Figure 5.

Reviewer #5

Updated Models

Comment: The code provides sufficient instructions for ad-hoc demultiplexing (after sequencing), and for live barcode balancing. The instructions and containers provided seem reasonable and well annotated for typical Nanopore users. However, training models provided are for RNA002 chemistries which have been deprecated and are not available anymore. Authors need to provide updated models OR provide a detailed README (along with associated tools and code) for users to train their own models using RNA004 chemistries which are currently available at Nanopore.

Response: We thank the reviewer for pointing this out. We have developed and validated models for the new RNA004 chemistry as outlined above and have updated the GitHub repository accordingly.

Reviewer #1 (Remarks to the Author):

Thanks to the authors for responding to my comments about warpDemuX.

As I indicated in my initial review, I am broadly in favor of the approach. However, as I have continued to review it I am struggling to clearly and simply identify the basic fundamentals in what the authors are saying.

We thank the reviewer for the positive feedback and we are happy to address the points raised and improve clarity accordingly. The point-by-point response can be found below.

Comment 1

I suggested the authors performed a barnyard experiment to determine the accuracy of barcode classification. This was not simply to determine if there is cross talk from sequences alone, but is to ensure the broader general accuracy of the method. If one is barcoding - and furthermore making decisions that influence sequencing behavior - then precise confirmation of the accuracy of barcoding and demultiplexing is essential. The RTA barcode shuffling approach may confirm this, but I struggled to find this clearly presented in the manuscript.

ANSWER: We appreciate the reviewer's suggestion regarding a barnyard experiment, which was noted in the initial revision as desirable in an 'ideal world' rather than an absolute requirement. We apologise if our motivation for experimental design wasn't entirely clear.

To rigorously validate our method's accuracy, we employed multiple complementary approaches that we believe provide even stronger evidence of our method's accuracy.

Importantly, our ground truth labels come from **in-line RNA barcodes** integrated into known RNA references using PCR. These barcode sequences were specifically designed for high discernability, enabling confident read assignment even in the presence of basecalling errors. We specifically chose this method over species-based genome mappings used in barnyard experiments due to several key limitations of the latter approach:

1. Mapping ambiguity:
 - a. Paralogous genes and repetitive regions often result in multi-mapping reads
 - b. Highly conserved regions between species can lead to cross-mapping
 - c. Incomplete or imperfect reference genomes affect mapping accuracy
2. Complex RNA processing dynamics:
 - a. Alternative splicing creates transcript variants that may not map uniquely to reference genomes
 - b. RNA editing and modifications can cause systematic mismatches with reference sequences

By shuffling the assignment of in-line RNA barcodes to RTA barcodes across replicates, we could assess classification accuracy under real experimental conditions, testing both technical aspects of barcode detection and biological variability inherent in RNA sequencing across RNA with varying poly(A) length and sequence contexts. Importantly, by having all barcodes present in all sequencing runs, sequencing batch effects are virtually eliminated, and our

measurements capture most (if not all) sources of error that may occur during multiplexed dRNA sequencing, including those not inherent to the algorithmic side, such as library prep.

Moreover, our method specifically processes only the RTA signal up to the polyA tail, deliberately disregarding downstream RNA transcript signals. The lack of correlation between RTA barcodes and RNA barcodes across replicates confirms that sequence variation downstream of the polyA tail does not affect demultiplexing accuracy. This is shown by the reproducible performance across independent test sets, for which the RNA barcode to RTA barcode assignments differed across data sets (Table 1, 2, SN3.1 and SN3.2 and Figure SN3.1). This independence from downstream sequence content, combined with our barcode-specific confidence score calibration (Figure SN 3.1, Figure SN 3.3, Figure SN 3.4, Table SN 3.1 and Table SN 3.2), provides robust validation of our demultiplexing accuracy.

The details of the RTA barcode shuffling approach are presented in Table S1 and Table S6, which detail the different assignments between RTA and (in-line) RNA barcode pairs across replicates. We apologize if this was not entirely clear in the revised manuscript and we changed the second revision accordingly to clarify the above mentioned points in line 87 - line 95.

Comment 2

I cannot see how the authors can make claims about the truth state of the SARS CoV-2 data they have presented. The exclusion of unclassified reads is an unusual rationale. These reads must have barcodes and so the lack of classification is a consequence of the method, not the input.

ANSWER: Thank you for raising this important point about unclassified reads and, we assume, its relation to the use of confidence score thresholds.

We want to clarify that **not all** signals classified as read by MinKNOW represent genuine RNA molecules. Our quality control process identifies various non-RNA signals, including electronic noise, pore blockages, and out-of-range measurements, which are appropriately left unclassified by design (assigned to the noise class).

For genuine RNA signals, we employ confidence score thresholds. Each classification is accompanied by a classification score, or confidence score (to be adjusted by a user, depending on the intended analysis and data). Setting a confidence score threshold below which reads are determined as unclassified is a standard part of classification algorithms. By focusing on high-quality classifications, we ensure reliable biological interpretation, as outlined below.

We chose a classification accuracy of 0.99, because high-quality classification is particularly crucial in **highly-imbalanced datasets** like our SARS-CoV-2 samples. For such extremely unbalanced input, the use of a high confidence score is inherently required to enable the analysis, because even a low misclassification rate from high-abundance samples could overwhelm the signal coming from low-abundance samples.

To illustrate this, consider a dataset with three samples of varying abundances: sample A and C (high abundance) and sample B (very low abundance), as depicted in the figure below. Even if the likelihood of reads from sample A being misclassified is low, if an inadequate confidence

threshold is chosen, the impact of these misclassifications varies dramatically depending on the target sample's abundance. While the absolute number of misclassified reads from sample A to samples B and C may be identical, their relative impact differs substantially. For the low-abundance sample B, these misclassified reads constitute a large proportion of its total signal (44%), potentially masking the true biological signal. However, for the high-abundance sample C, the same number of misclassified reads represents only a small fraction of its total signal (4.8%), having minimal impact on downstream analysis. This demonstrates that the impact of misclassification must be considered in the context of the target sample's abundance. By enforcing an adequately high confidence threshold, the spillover can be minimized, ensuring reliable biological interpretation across all abundance levels.

In our SARS-CoV-2 samples, the supernatant samples could be regarded as 'sample B' groups. Our selected high confidence threshold reflects this biological reality.

We further elaborated on our justification of the chosen confidence threshold (line 158-162), and included the illustrative example in Section SN.4 (subsection WarpDemuX confidence threshold choice).

Comment 3

Figure 4f is puzzling. What is the control group here? Is it the whole of the no WDX portion of the flowcell? Or is it the no balancing component? To properly evaluate the comparison these details need to be recorded. The results are not shown for the read count or the base count. I presume the comparison must be against the no balancing section. Please clarify.

ANSWER: We thank the reviewer for this important question about the experimental design of Figure 4f. We apologize for any lack of clarity in our presentation. The direct control group consists of the 40 channels without barcode balancing enabled ('no balancing' portion in Figure 4b), as correctly stated by the reviewer. This split-flow cell design ensures direct comparison between balanced and unbalanced sequencing under identical experimental conditions. The additional 312 no-WDX channels served as internal quality control to compare

data throughput with and without the WarpDemuX adapter detection step. We have clarified this experimental design in the figure and the legend.

With regards to the reviewer's remark on the missing results for the read count or the base count balancing strategies, we thank the reviewer for their deep interest. While we had already included the relative counts per barcode across balancing strategies in Figure SN 5.3, our supplement did not include absolute read count results akin to Figure 4f. We have now added the new Supplementary Figure SN 5.4, in which we also present the data shown in Fig. 4f for the other two balancing strategies (see also comment 8).

Comment 4

The authors have modified the text about the issues around base calling and signal processing, but this text is still misleading.

Comment 4a

The authors state that GPU is computationally expensive. An option that doesn't require GPU simply doesn't use GPU. If using Nanopore sequencing one needs to have GPU to enable base calling.

Comment 4b

Similarly the argument about the number of reads that can be processed per second is misleading. WarpDemuX can process a significant number of reads - 50,000 per minute. But, a MinION running at full speed on the assumption of 0.1 seconds of data per read can generate at max 5,120 reads. Thus the 10,000 reads/minute of SeqTagger is still more than sufficient. For a PromethION device, this isn't the case. But it is unclear if a promethION device can supply signal in chunks of 0.1 seconds.

ANSWER: Whilst we appreciate the reviewer's perspective, we disagree that the text is misleading.

In the absence of a specific reference, we assume that in comment 4a and 4b, the reviewer is referring to his/her initial comment for which we added the sentence "..." (line 275 in the first revision). *For RNA004, the authors present performance metrics on a 4-barcode model, which are comparable with WDX4 performance, yet require significant computational overhead through GPU acceleration while being five times slower, processing approximately 10,000 reads/minute compared to WDX4's 50,000 reads/minute (Fig. 5g).*

First, we would like to clarify that WarpDemuX supports both live-demultiplexing and post-acquisition demultiplexing. In post-acquisition demultiplexing, demultiplexing is run on data from an ongoing or completed sequencing run, as such, there is no barcode balancing or read rejection taking place. For post-acquisition demultiplexing, the processing speed of the demultiplexing algorithm can still be significant, as it impacts the maximal throughput and turnaround time of direct RNA sequencing analysis.

Second, we want to reiterate that live (barcode balancing) demultiplexing and post-acquisition demultiplexing require different performance metrics. While in both cases it is important to

have enough throughput to keep up with sequencing acquisition speed, for barcode balancing demultiplexing the latency is an additional critical parameter. That is, for each read, the time from signal acquisition to response needs to be as short as possible. We elaborate on this further in Comment 6, where a similar point about basecaller-based adaptive sampling approaches is raised.

Reply to 4a

When performing GPU-accelerated basecalling, the GPU is most often the limiting resource. For example, Oxford Nanopore Technologies recommends a “high-end Nvidia GPU (RTX 4090 recommended)” for their P2 Promethion device due to GPU computational demands of basecalling (<https://nanoporetech.com/document/requirements/promethion-2s-it-req#configuring-a-new-computer>, accessed January 10th). Thus, a second process competing for those GPU resources can be legitimately called a substantial computational cost. While in resource-rich environments, multiple GPUs could be employed to perform basecalling and GPU-demultiplexing in parallel, in resource-constrained settings, these must run sequentially, extending analysis time. In contrast, a CPU-based approach can run alongside GPU-basecalling without adding to the total analysis time.

While we agree that GPUs are *de facto* required for basecalling in nanopore sequencing, they are not *strictly* required (see the official documentation on CPU-based basecalling, <https://nanoporetech.com/platform/technology/basecalling>, accessed January 10th). Moreover, basecaller-independent nanopore analysis is an active area of research, for example in pathogen monitoring. Finally, CPU-based solutions in bioinformatics, especially those runnable on a personal laptop, can be essential in resource-constrained settings, which many biology labs still face around the world even today.

We have updated our statement on the computational overhead (line 276-281) by expanding that our claim for substantially more compute cost is based on the 5x higher throughput of WarpDemuX with reduced computational requirements, that is, a laptop CPU compared to a mid-range professional desktop GPU.

Reply to 4b

We thank the reviewer for this astute observation about real-time processing requirements.

While a throughput of 10,000 reads/minute might suffice for live-demultiplexing on a single MinION flow cell, we would like to clarify that SeqTagger was not benchmarked with live demultiplexing, where throughput is likely substantially lower in exchange for achieving reduced latency (see Comment 5). The processing speeds discussed in line 275 are in the context of post-run demultiplexing.

In this post-run scenario, WarpDemuX's superior processing speed (50,000 reads/minute) significantly reduces analysis time - this is particularly important for high-throughput devices like PromethION, where the time savings can be substantial.

As an example for estimating the effect of different demultiplexing solutions on turnaround time, we estimate the yield of a single Promethion flow cell to be 1 million reads per hour (i.e. 16,7k reads per minute). While the laptop CPU used for WarpDemuX benchmarking would be able to keep up with three Promethion flow cells, the RTX A4000 GPU used in SeqTagger

would not keep up with a promethion flow cell, and take an additional 60% of the sequencing time to finish demultiplexing. Such a delay may not be significant in research labs, where sequencing is rarely performed at capacity, however other settings where more than one Promethion flow cell are run in parallel will certainly benefit from substantially reduced turnaround times.

We have clarified the fact that SeqTagger is a post-acquisition demultiplexing method (line 277).

Comment 5

This also raises a false equivalence with the comparisons made between basecalling and signal analysis. In this work the authors have reduced the `break_read_seconds` parameter to 0.1 seconds. Prior work changed this to 0.4 seconds for base calling. The standard MinKNOW setting is 1 second. The revised text the authors present is incorrect.

ANSWER:

With regards to comparisons between basecalling and signal analysis, we would like to refer to our answer to Comment 6.

Further, it is unclear to us which part of the text the reviewer deems incorrect. To clear up a potential misunderstanding, our empirical optimization did not merely lie in us reducing the `break_reads_after_seconds` time to 0.1 seconds. Rather, this change was introduced before we started our empirical optimizations. Specifically, these changes involved the introduction of new read classification parameters ('strand_low_range', 'stalled', 'polya' with specific signal metric thresholds) and additional channel states ('polya', 'stalled', 'long_stalled', 'blocked' with classification regex patterns), as displayed in the `minknow_config` folder of our github.

We would additionally like to point out that the 0.4 seconds `break_reads_after_seconds` parameter has been established by Payne et al. for DNA sequencing, not RNA sequencing. While 0.4 seconds is successful for DNA sequencing, our testing revealed that this setting on its own led to misclassification of pore states in RNA sequencing, particularly affecting the detection of short stalls versus blockages. Our further optimized 0.1-second sequencing protocol thus required the additional alterations mentioned above, and represents a careful balance between rapid decision-making and maintaining data quality in the context of RNA sequencing. We have included the details on our optimization process in Supplementary Note 7 (Reducing the Chunk Size), and included detailed results of our optimization analysis as part of our revision in Figures SN 7.2 and SN 7.3.

Comment 6

(Line 199-207) "Although basecalling approaches effectively determine nucleotide sequences, they impose significant computational overhead and require the accumulation of a sizable amount of raw signal data before analysis can commence. This inherent latency introduces delays in adaptive sampling decision-making and allows unwanted molecules to traverse further through the pore before rejection. This extended pore occupancy negatively impacts pore lifetime and sequencing yield, reducing the effectiveness of the approach

(Kovaka et al.). Signal-based approaches circumvent these limitations and can enable more rapid molecule rejection, preserving pore integrity and resulting in better overall enrichment and/or yield of low abundance samples. Yet, this barcode balancing approach remains unexplored in the context of dRNA-seq.” Basecalling works on as little as 40 bases of DNA - which equates to 0.1 seconds. To make the statements above that the authors claim they would need to test base calling performance at 0.1 second chunks. The inherent latency is not derived from basecalling - it is derived from the underlying chunk size switch.

ANSWER: We thank the reviewer for this important technical discussion.

Before diving into the remarks on latency, we would like to point out an important difference between DNA and RNA nanopore sequencing in this context: namely, the fact that RNA has a lower translocation speed compared to DNA. Therefore, while 0.1s equates to 40 DNA bases (R10.4.1, 400b/s), for RNA, 0.1s equates to 7 (RNA002, 70b/s) or 13 (RNA004, 130b/s) bases.

Then, to clarify our point: what we mean by latency is the time delay between initiating an action and receiving a response. Latency includes not just processing time (how long it takes to perform computations or execute instructions), but also data transmission time, time spent waiting in queues, time spent reading from or writing to storage and any other delays in the system. A system can have fast processing speed but still experience high latency.

We would like to clarify that “accumulation of a sizable amount of raw data” in line 199-207 relates to the fact that the deep learning architectures used for basecalling operate in batches, meaning generally more than one read (or alternatively multiple chunks of a read) are basecalled in parallel. This batching of signals introduces inherent latency.

We have revised the text to better reflect the nuance of batch processing: (line 203-205) *“Although basecalling approaches effectively determine nucleotide sequences, the models used for basecalling require batching, which requires the accumulation of a sizable amount of raw signal data before analysis can commence. “*

If we take a broader look at the difference between WarpDemuX and GPU-based approaches for adaptive sampling, we have the following trade-offs:

1. WarpDemuX (CPU, Lightweight, Single-read Processing):
 - a. Processing Speed: Lower raw computational throughput compared to GPU
 - b. Latency: Very low, since each input is processed immediately.
 - Each new input → immediate processing → immediate output
 - No waiting time for batch accumulation
2. GPU (Heavy ML Model, Batch Processing):
 - a. Processing Speed: Much higher overall throughput, can process many inputs simultaneously
 - b. Latency: Higher, due to several factors:
 - Need to wait for enough inputs to form a batch
 - Time to transfer data to GPU memory
 - More complex model may take longer to produce outputs
 - Time to transfer results back from GPU

While GPU-based solutions can achieve high throughput through simultaneous processing of many inputs, signal batching, data transfer to and from the GPU device and spinning up of computational kernels all contribute to latency. Our CPU-based approach demonstrates a proof-of-concept for direct RNA live barcode balancing that minimizes these overhead costs.

Nevertheless, we agree with the reviewer that including supporting data for these concepts would further strengthen the manuscript and we appreciate the reviewer's request to explicitly examine basecalling latency. We have therefore estimated basecalling latency with reduced chunk size, as requested.

In our analysis, we measured basecalling latency by replaying a run that matched the settings used for WarpDemuX adaptive sampling (0.1 second chunk size, 500-4000 samples submitted to classification). In order to evaluate the computational overhead we tested basecaller adaptive sampling latency in three configurations: 1) adaptive sampling with the fast model, with disabled live basecalling (best case for adaptive sampling), 2) adaptive sampling with the high accuracy (hac) model, with disabled live basecalling (increased accuracy adaptive sampling), and 3) adaptive sampling with the high accuracy model with live basecalling with high accuracy (hac) model (most similar in accuracy to live WarpDemuX). The tests were performed on a user-grade system (Ryzen 7 5800X3D CPU, 32GB RAM, RTX3070 GPU) and revealed that:

1. The latency of adaptive sampling basecalling with the fast (low accuracy) model consistently was around 400 and 500 milliseconds, on Linux and Windows respectively, regardless of the number of concurrently based called reads (1-30 reads).
2. The latency of adaptive sampling basecalling with the hac (high accuracy) model was 800 ms, revealing a considerable amount of compute time can be spent on basecalling if high accuracy is warranted
3. Concurrent high accuracy adaptive sampling with high accuracy live basecalling resulted in latency of 1000 ms, highlighting significant computational overhead of basecalling
4. Note that this latency only includes sequence determination (basecalling) and excludes the necessary barcode assignment through alignment, balancing decisions and the latency of the rejection request itself.

These results align with ONT's own documentation regarding the computational demands of basecalling-based adaptive sampling. As stated in their documentation (<https://nanoporetech.com/document/adaptive-sampling>, accessed Jan 9th 2025): "*Adaptive sampling requires a lot of computing power because of its need to basecall, align, and make a decision on all the strands captured in real-time. Running live basecalling during an adaptive sampling run may lead to reduced enrichment (reduction in on-target coverage obtained) due to the lack of resources to handle both basecallers.*"

Furthermore, ONT's data, included under section '12. Device specifications' on the same page, shows that in DNA sequencing (where adaptive sampling is currently available for fast basecalling at 1s chunks), the N50 length at rejection is 679 bases. Given DNA's translocation speed of 400 bases/second and a mandatory acquisition of one chunk, this indicates an inherent system latency of approximately 0.7 seconds $((679-400)/400)$ for the complete process of basecalling, alignment and decision-making.

Two important considerations emerge from our analysis:

1. **Latency Impact:** Basecaller-based adaptive sampling exhibits substantially higher latency (400-1000 milliseconds) compared to our signal-based approach (median 43 milliseconds). This 9-23-fold difference in latency directly affects enrichment efficiency, as with longer latency target molecules occupy nanopores longer before decisions can be made.
2. **Resource Requirements:** Basecalling demands substantial GPU resources, and live basecalling can affect adaptive sampling performance. As confirmed by our empirical data as well as ONT's documentation, this often necessitates either additional GPU hardware or disabling live basecalling to maintain performance. In contrast, WarpDemuX's signal-based approach operates efficiently without GPU requirements.

In conclusion, the presented findings support our original assertion that direct raw-signal processing offers advantages over basecalling-based approaches in particular for real-time applications like adaptive sampling. While we acknowledge that further accelerations could theoretically be achieved by optimized basecalling approaches, we trust the reviewer can appreciate that in this work we did not aim to make direct comparisons to such theoretical alternatives.

We have included the discussed analysis in SN 8 Appendix: Analysis of Basecalling-based Adaptive Sampling Latency.

Comment 7

The authors quote Kovaka et al on impacts on pore lifetime, but this is again linked to the *change in unblock time* and nothing else. The signal-based approach accelerations that are reported here are equally achievable with basecalling approaches. After all, the basecalling method is itself a signal based approach. The authors do not demonstrate any of the claims they make about pore integrity, better overall enrichment or yield of low abundance samples. *I do not disagree with the authors point that processing less data faster will result in better performance. I do disagree that this is an inherent property of the method presented here.*

ANSWER: While we appreciate the reviewer's careful analysis of our claims regarding pore lifetime and enrichment performance, we disagree that we 'do not demonstrate any of the claims [we] make'.

We already elaborated above the advantage of lower latency and reduced computational requirements of our method to basecaller-based approaches in response to comment 6 above.

We agree with the reviewer that the improvements in sequencing efficiency are related to unblock times, that is: the magnitude of the method's latency, including the speed of processing, and the cooldown time between unblocking requests. While we agree that similar processing speed improvements could theoretically be achieved through other methods, such as optimized basecalling approaches, we trust that the reviewer appreciates that our approach provides an actual, validated solution that works effectively with current nanopore technology.

Regarding the demonstration of our claims on pore integrity, enrichment and yield of low abundance samples, we provide several pieces of concrete evidence in the manuscript:

1. For pore integrity, we refer to the Figure SN5.4 and the accompanying text: "*Ultra-fast barcode classification with adaptive-sampling can lead to rejection frequencies above 50% (Figure 4e), depending on the abundances of the multiplexed samples and balancing strategy. An unsuccessful rejection may result in pore blockage, reducing the lifetime of the pore and sequencing yield of the run. We aimed to minimize the number of pore blockages upon unsuccessful rejection by rejecting reads while they are still within the poly(A) tail. We analyzed the effect of different balancing strategies on pore lifetime, and found no substantial differences between pores with and without adaptive sampling, regardless of the strategy used (Fig. SN.5.4)*".
2. For better overall enrichment, we would like to point out Figure 4e, which shows an increase in the number of reads observed per pore, independent of the balancing strategy.
3. For enrichment of low abundance samples, Figure 4f directly demonstrates successful enrichment of low-abundance viral RNA, achieving 33-73% increased recovery compared to non-balanced channels.

To improve the clarity of the text with regards to the relation between processing speed and performance we have made the following changes:

Previous version (Line 203-207): "*This extended pore occupancy negatively impacts pore lifetime and sequencing yield, reducing the effectiveness of the approach [20]. Signal-based approaches circumvent these limitations and can enable more rapid molecule rejection, preserving pore integrity and resulting in better overall enrichment and/or yield of low abundance samples. Yet, this barcode balancing approach remains unexplored in the context of dRNA-seq.*"

Revised version (Line 207-210): "*This extended pore occupancy negatively impacts pore lifetime and sequencing yield, reducing the effectiveness of the approach [20]. Reducing inherent latency can enable more rapid molecule rejection to preserve pore integrity, resulting in better overall enrichment and/or yield of low abundance samples.*"

Comment 8

I would strongly suggest the authors consider reducing the complexity of the supplementary material and increasing the transparency of reporting. Figure SN.5.3 shows the percentage of reads rather than absolute numbers- absolute numbers would be a useful comparison.

ANSWER: We thank the reviewer for this constructive feedback about data presentation. After careful consideration and with particular focus on transparency and in regards to the technical comments made by the reviewer, we believe that the information currently included in the supplementary material is non-redundant, instructive for reproducing the results and required to inform further optimization.

Our extensive supplementary material aims to provide complete transparency in our methodology and analysis. Regarding Figure SN.5.3, we chose to present percentages to clearly illustrate the relative effects of different balancing strategies and the difference between

count- and base-based balancing. However, we agree with the reviewer that reporting absolute numbers could provide further insight. We have now included these in the new Figure SN 5.4 (also see our reply to Comment 3).

Reviewer #2 (Remarks to the Author):

I have carefully reviewed the revised manuscript and responses to the reviewers' comments. I am pleased to see how thoroughly and diligently the authors have addressed all the feedback. The incorporation of RNA004 kit support significantly enhances the manuscript's relevance for current sequencing technologies. The authors' enhanced analysis of potential biases, expanded discussions, and additional validation data have substantially improved the scientific rigor of the work. The revised figures and supplementary materials provide excellent clarity to the work.

After thorough consideration, I believe the manuscript in its current form is suitable for publication.

We thank the reviewer for the positive feedback and the appreciation of our work.

Reviewer #3 (Remarks to the Author):

I am pleased by the authors' modifications and changes to their manuscript.

Reviewer #3 (Remarks on code availability):

I could install the code and the test data have been provided this time.

The GitHub page is also well prepared and enables a new user to get familiar with the software.

We thank the reviewer for the positive feedback and the appreciation of our work.

Reviewer #4 (Remarks to the Author):

In the revised manuscript, the Authors now provide evaluation of WarpDemuX and barcodes using the current generation nanosequencing chemistry and have added a test example to the GitHub repository to demonstrate the expected input and output files. Figures have been improved. I am satisfied with the Authors' additions to the revised manuscript. The methodology could be useful for the field in the current form.

We thank the reviewer for the positive feedback and the appreciation of our work.

Reviewer #1

I thank the authors taking the time to address my comments.

We sincerely thank the reviewer for their thorough and constructive feedback.

Comment 1.

The authors have explained the shuffling clearly enough for me to follow. The text changes are also clear in this regard.

We are happy that the reviewer is content with our revisions.

Comment 2.

I agree with the authors that not all signals initially classified by MinKNOW represent genuine RNA molecules. I also agree with the issues of the difficulties of classification when you have a highly imbalanced data set. Of course setting a high threshold is important. My question here is why are the unclassified reads not explored. For example, of the 363,283 reads that pass a Q score filter, 16592 reads do not have barcodes. Is that because they are completely missing - i.e they have no barcode - or is the barcode there, but not detectable by warpDemux?

The reads the reviewer highlights are reads that pass the basecaller Q score filter, but do not pass internal adapter signal quality checks preceding demultiplexing (WDX QC), or were predicted as the noise-class. WDX quality checks are handled by the tool ADAPTEd and entail signal characteristics validation including poly(A) detection. We have now clarified this in the text on lines 157 (results) and 440-441 (methods). Since poly(A) tails are inherently present in all reads due to being a fundamental requirement for standard dRNA-seq library preparation, they serve as a reliable internal validation element. When poly(A) detection fails, it may indicate poor signal quality at the adapter region, preventing reliable demultiplexing. We note that in ongoing work, we are developing methods to extend our approach to non-polyadenylated RNA species such as tRNA and rRNA, which would broaden the applicability of our method beyond poly(A)-enriched transcripts.

The reads that are subsequently filtered out (109,739 reads) all have barcodes but - as pointed out by the authors - have a lower classification confidence. The authors state (lines 163-166) that the correlation between barcodes and EGFP sequences is effectively perfect in the reads that pass classification. Does this correlation remain in the 109,000 reads that can be assigned a barcode but which - quite rightly - may be excluded from the subsequent analysis for other reasons.

The strong correlation between barcode and EGFP sequences diminishes when low-confidence predictions are included in the analysis, which aligned with our expectations. We have now also included these observations in the text (line 943-945): *“When low-confidence predictions were included, this clear sample separation deteriorated, confirming our hypothesis about the importance of stringent thresholds.”*

Line 160 is missing a specific reference to a figure.

We had erroneously included the empty reference, it has now been removed. Thank you.

Comment 3

Thanks to the authors for providing extra detail. The new figure presented in supplementary figure 5.3 is helpful, but is not referenced in the main text.

We appreciate the reviewer's attention to detail. We would like to kindly clarify that the figure in question was actually Fig. SN5.4 (now: Fig. SN5.5), which was added in the last revision, rather than Fig. SN5.3 (now: Fig. SN5.4). Fig SN 5.4 (now: Fig. SN5.5) is referenced in the main text in line 230-231. For completeness, we should mention that Fig. SN5.3 (now: Fig. SN5.4) serves to support the text in Supplementary Notes 5 (SN5), which is referenced in the main text at lines 220, 291 and 491.

I presume that by "passing" read counts and "passing" read bases the authors are referring to the data that passes the thresholds already discussed. The real question is if the method results in more absolute data on target. The relative proportions shown in SN.5.3 a and b show that the data are normalized, but do not show the absolute enrichment. This is shown in SN 5.4. The figure I was hoping to see was a panel similar to SN.5.4 but with the total bases sequenced (absolute number of bases barcode - not relative numbers) as well as a count of the total number of reads and bases unclassified in each balancing strategy. This would unambiguously demonstrate the enrichment and removes any ambiguity from the data.

We thank the reviewer for this suggestion. To clarify, "passing" reads in the figure refers to non-rejected reads that are written out to the pod5_pass folder during sequencing. We have clarified this in the figure caption. We have also added a new table (Supplementary Table SN.5.1) showing the absolute numbers of bases and reads for each balancing strategy, as requested by the reviewer.

Comment 4 (a).

The authors fail to mention that the recommended GPU compute - the RTX4090 - is sufficient for both real time base calling and implementation of adaptive sampling with respect to DNA. Given the higher sequencing speed of DNA sequencing to RNA it is ergo simpler to run the analysis on a GPU. It is also not correct to assume that CPU resource is unlimited in these scenarios. It is very easy to overwhelm the CPU requirements whilst running live code alongside minKNOW. The only reference I can find to compute resource is an Intel i7-1165G which is the minimum requirement for running a minION. This is good performance, but is this the set up the authors used for their experiment?

I completely agree with the authors that efficient and effective code is vital. I disagree that there is evidence to say that this is a practical limitation.

While we acknowledge that both CPUs and GPUs can indeed be pushed to their operational limits, we did not encounter CPU constraints when running WarpDemuX. This stands in contrast to our RNA basecalling-based adaptive sampling work (detailed in SN8), where we did observe GPU resource limitations. We have specified the system specifications used for our barcode-specific adaptive sampling in the newly added Supplementary Materials 7 (SM7), which is referenced in the text in lines 493-494. We apologize for the oversight and thank the reviewer for bringing it to our attention.

Comment 4 (b)

The post-run scenario the authors describe is not really likely. I come back to my point that GPU base calling is - really - an essential pre-requisite to any laboratory running nanopore sequencing at scale. A laboratory using multiple promethION flowcells is likely to be able to implement GPU calling. Nonetheless I do take the point that post run demultiplexing rapidly is good.

We agree with the reviewer's assessment regarding GPU basecalling for large-scale operations while maintaining that our method offers benefits for both large and smaller-scale applications.

Comment 5

Re-optimising the sequencing toml files and configuring these for short chunks is a lot of work. I don't understand the comment at line 1100 in the supplementary material - "Unfortunately, these necessary changes resulted in performance highly similar to the default protocol" - why is this unfortunate? In figures sn 7.2 and 7.3 are all of these data tuned on simulated runs (playback) or did the authors try switching between protocols on a real sequencing run to determine no negative effects. The requirement to run a modified sequencing script is something that should be made very clear in the main text? I cannot find this anywhere?

The word 'Unfortunately' in line 1100 refers to a key tradeoff in protocol optimization. While our initial modifications achieved ~30% better yield in simulated runs, these changes compromised the ability to detect and unblock stalled pores during actual sequencing. After implementing additional changes to maintain pore unblocking functionality in real runs (specifically, the new 'stalled' state classification), the performance metrics of our custom protocol—including read count, length, and accuracy—became comparable to those of the default protocol, as shown in Fig SN.7.2. We have clarified this in the text in line 1113-1115.

We confirm that we have indeed validated the incremental changes to the protocol on both simulated and real runs.

Regarding protocol switching, the reviewer is correct that the instructions for running a modified sequencing configuration should be clearly communicated. We chose to maintain up-to-date installation and setup instructions in our GitHub repository due to the evolving nature of MinKNOW updates from Oxford Nanopore Technologies. We have now added an explicit statement in the Methods section that explains that these software configuration instructions are included in our GitHub repository (lines 492-493).

My comment about the text being misleading is that - for adaptive sampling - the fact that the break reads parameter does not have to be 1 has been previously reported. If the authors have resolved the toml files for this then have they tested the performance of inbuilt adaptive sampling using base calling against their method? I do realise that this is almost certainly impossible in the context of this specific experiment as the integration of the barcode into the adapter precludes this. But it would be of interest to know if the performance improvements translate. In addition, is the base calling of the data impacted in anyway by the change in sequencing parameters - one presumes not?

As the reviewer correctly points out, this comparison is not possible in the context of this experiment. We confirm that the sequencing parameters (MinKNOW configuration) have no effect on the signal trace and thus do not impact basecalling.

Comment 6

The authors have made a number of assumptions about how the base calling process is configured and how “batching” occurs. The models used do not require batching - reads can be sent as singletons or as multiples. The authors have no doubt noticed that the reads they receive via the read until API are themselves in batches. My point was that the authors could test performance of base calling at 0.1 seconds and determine if the improvements come from their optimized algorithms or simply theft that data are now available in 0.1 second bursts. In the authors comparison of the approaches they state the latency is low since each read is immediately processed. The same can be achieved with base calling. So - the claim the authors make with respect to latency and the requirement for waiting for enough reads to batch is incorrect. A single read can be sent (and is sent further down).

Our comparison focused on practical implementations where batch processing is commonly used to optimize GPU utilization. While single-read processing is technically feasible, it reduces the efficiency gains offered by GPU parallelization. This is also shown in Fig SN 8.1, where the latency of basecalling for RNA adaptive sampling at 0.1s data chunks at a batch size of 1 is comparable to the latency of basecalling 30 reads simultaneously. We have clarified this nuance of batching in the text in lines 205-206.

The authors have addressed my concerns with respect to running some simulations, but they do so within the framework of readfish which, as I understand it, inherently batches reads rather than sending them asynchronously. The authors state their own model calls one read every 43 Ms. I couldn't immediately find the data for that but I do not disbelieve it. It would be worthwhile reporting the mean time as well as the median.

We have now included both the mean and median processing times in line 1239 of the revised manuscript. During this revision, we corrected a minor error: the median WDX Adaptive Sampling (AS) time is 41ms rather than 43ms. The statistics on WDX AS durations are further detailed in SN5, subsection “Efficiency of WarpDemuX adaptive-sampling”, which we have now referenced in line 1227. For completeness, we also added a new Supplementary Figure SN 5.3 displaying the distribution of AS analysis durations, showing the mean is heavily influenced by a low number of outliers.

The authors test more complex models (hac etc) and I agree that the performance drops here. The authors state they test HAC as this gives equivalent performance to WarpDemuX - how is this shown?

During WDX Adaptive Sampling (AS), RNA transcripts from AS-accepted reads underwent live HAC basecalling, simultaneously with the sequencing. Therefore, AS basecalling with simultaneous HAC basecalling of accepted reads represents the most directly comparable scenario to WDX's operational pipeline. We have clarified this experimental design and the rationale behind our comparison (line 1187-1189).

The authors quote ONT's notes that say that necessary compute is required for adaptive sampling to run successfully. I agree with this. I also agree with the shown figures that ONT's N50 at rejection is around 680 bases. And I agree with the authors calculation of the approximate latency in the system (0.7 seconds) after data collection.

We appreciate the reviewers' confirmation.

What are the equivalent values for the code in this work running on an entire flowcell (not a small section thereof). What are the values when running a similar base calling experiment using the 0.1 second unblock time?

We anticipate linear scaling with the number of pores used. However, we did not perform this experiment.

I note that the authors state "While we acknowledge that further accelerations could theoretically be achieved by optimized basecalling approaches, we trust the reviewer can appreciate that in this work we did not aim to make direct comparisons to such theoretical alternatives.". Indeed - I agree with this. But it is the authors who are not acknowledging this in the manuscript text.

We have now included the following acknowledgement (lines 1235-1236): "In the future, further optimization of basecalling approaches for real-time applications may address these challenges."

I really think the approach developed by the authors is innovative and of interest. I note the comment in line 305 of the manuscript that the approach is independent of base calling. This is a really neat trick, but as the authors acknowledge requires new SVM-DTWD kernels if the chemistry changes. If one had an RNA barcode that could be base called, how would this compare?

Updates to the chemistry require retraining, independent of the inference algorithm used. If one had a RNA barcode that could be base called, updates to the chemistry would likewise necessitate retraining the basecaller (taken care of by ONT) for the barcode to be basecalled correctly. We discuss the topic of basecallable RNA barcodes in line 51-59.

Comment 7.

The authors point that they aim to unblock reads whilst in the poly(A) tail reduces pore blocking is well shown. They also indicate their method has no negative impact on pore life span. These data are excellent. The issue with the base calling approach is more nuanced. If the authors wish to test this they should unblock every read on a flow cell (using base calling) and look at the loss of pores compared to a control region. The difference is imperceptible and only really emerges 24 hours into a run. I do not know how long it takes on RNA. I would imagine that RNA might be more sensitive to blocking - but the authors do not show this. The authors have modified their statement, but really the point here is that they do not show the impact of base calling adaptive sampling on RNA sequencing performance and so they cannot make a comparison.

We appreciate the reviewer's recognition of our unblocking data and thank the reviewer for the appreciation of our additional experiments. We have focused our manuscript on demonstrating the effectiveness of our basecaller-free approach. We do not study base-calling based adaptive sampling in the manuscript and did not aim to make a direct comparison to barcode-basecalling adaptive sampling, because it doesn't exist yet for direct RNA sequencing.

Our revised statement was meant as a general statement, not a direct comparison. We have revised the statement and the preceding section once more to better reflect this nuance (line 206-212).

Comment 8

I am sorry it has taken a long time to respond to the authors. This is - in part - because the supplementary notes, explanations and figures are now at 65 pages. Much of this is to support the authors relative comparison claims.

As I understand it, the approach that the authors present cannot use base calling. So it isn't an option. The solution is elegant and works well - providing a good and efficient RNA barcoding approach. This is what the manuscript should be emphasizing (in my view). The adaptive sampling data shown in figure sn 5.4 show a less than 2 fold improvement in the number of reads from the least abundant barcode from an awful lot of work. In my view this is the least exciting aspect of the paper - far more exciting is the ability to efficiently barcode RNA samples.

We thank the reviewer for this perspective. While we agree that the efficient RNA barcoding approach is a key strength of our method, the adaptive sampling results demonstrate an important proof-of-principle for barcode-based enrichment. The observed two-fold enrichment, while modest, should be viewed in historical context: early implementations of DNA adaptive sampling also showed limited enrichment compared to current capabilities. Similarly, recent work on direct RNA adaptive sampling using basecalling of the RNA transcript has demonstrated lower enrichment factors than we show here (Wang, J. *et al.*, Nat Commun (2024): 22-30%). It's important to note that enrichment efficiency typically improves for lower abundance targets - a pattern observed across adaptive sampling applications. While our current enrichment levels may not drive immediate widespread adoption (similar to early DNA barcode balancing applications), they represent a meaningful improvement that could enable otherwise challenging downstream analyses, particularly for low abundance transcripts. We view these results as an encouraging first step, with potential for optimization and improvement similar to the evolution seen in DNA adaptive sampling methods.

Reviewer #1 (Remarks on code availability):

I have inspected the code but I have not been able to run it as it requires generation of a sequencing library to test properly.

We would like to clarify that sequencing library generation is only required for executing real adaptive sampling runs. The functionalities of the adaptive sampling software, as well as the core barcoding functionalities can be tested independently.

Reviewer #1

The authors have largely addressed many of my comments.

We thank the Reviewer for their continued careful review of our manuscript.

The authors incorrectly state that base calling approaches to RNA adaptive sampling are not available - for example see Wang, J., Yang, L., Cheng, A. et al. Direct RNA sequencing coupled with adaptive sampling enriches RNAs of interest in the transcriptome. Nat Commun 15, 481 (2024).

<https://doi.org/10.1038/s41467-023-44656-3>

We appreciate the reviewer pointing out the work by Wang et al. (2024) on RNA adaptive sampling. To clarify, we stated our work demonstrates proof-of-principle for barcode-based adaptive sampling, which is different from sequence-based adaptive sampling approaches. While sequence-based methods like Wang et al.'s are valuable for targeting specific transcripts, they cannot distinguish between samples from the same species under different conditions (i.e. as in our SARS-CoV-2 example). Our barcode-based approach enables such sample-specific enrichment. We acknowledge the work of Wang in our manuscript and in our previous response letter (cited as Ref. 19).

In my view the only way to really test adaptive sampling experiments is by running a real experiment on a sequencing flow cell. Whilst one can simulate with playback and other simulations it is not the same as running an actual experiment. Similarly, it is not correct to assume that performance will scale linearly (presumably inversely) with pore count - as MinKNOW itself is adding complexity to the process.

We agree that the best way to test adaptive sampling experiments is to run a real experiment on a sequencing flow cell. We have performed our adaptive sampling experiments on actual flow cells, rather than relying on simulations or playback data. The Reviewer raises an important point about the complexity that MinKNOW *may* add to the process, and we agree that performance scaling may not be strictly linear with pore count due to MinKNOW. However, the design and performance of MinKNOW is beyond our control (as it is developed by ONT).

I remain of the view that the true innovation here is the excellent barcoding approach and that the benefits of adaptive sampling on RNA are minimal.

We thank the Reviewer again for their valuable feedback throughout the review process.